# Regularized Adaptive Momentum Dual Averaging with an Efficient Inexact Subproblem Solver for Training Structured Neural Network

**Zih-Syuan Huang**
Department of Computer Science and Information Engineering
National Taiwan University
Taipei 106, Taiwan
r11922210@ntu.edu.tw

**Ching-pei Lee**
Department of Advanced Data Science
Institute of Statistical Mathematics
Tachikawa, Tokyo 190-8562, Japan
chingpei@ism.ac.jp

## Abstract

We propose a Regularized Adaptive Momentum Dual Averaging (RAMDA) algorithm for training structured neural networks. Similar to existing regularized adaptive methods, the subproblem for computing the update direction of RAMDA involves a nonsmooth regularizer and a diagonal preconditioner, and therefore does not possess a closed-form solution in general. We thus also carefully devise an implementable inexactness condition that retains convergence guarantees similar to the exact versions, and propose a companion efficient solver for the subproblems of both RAMDA and existing methods to make them practically feasible. We leverage the theory of manifold identification in variational analysis to show that, even in the presence of such inexactness, the iterates of RAMDA attain the ideal structure induced by the regularizer at the stationary point of asymptotic convergence. This structure is locally optimal near the point of convergence, so RAMDA is guaranteed to obtain the best structure possible among all methods converging to the same point, making it the first regularized adaptive method outputting models that possess outstanding predictive performance while being (locally) optimally structured. Extensive numerical experiments in large-scale modern computer vision, language modeling, and speech tasks show that the proposed RAMDA is efficient and consistently outperforms state of the art for training structured neural network. Implementation of our algorithm is available at https://www.github.com/ismoptgroup/RAMDA/.

## 1 Introduction

Since the recent emergence of ChatGPT, large language models (LLMs) and other huge deep learning models have garnered much attention and popularity, even among the public who are unfamiliar with machine learning. A challenge with such gigantic neural network models is their vast number of model parameters, reaching hundreds of billions, resulting in expensive storage and inference. It thus becomes crucial to find ways to exploit structures in

38th Conference on Neural Information Processing Systems (NeurIPS 2024).

trained models to reduce their spatial and prediction costs without degrading the prediction performance. An active line of research is to explicitly add a nonsmooth regularization term to the training objective function and apply proximal stochastic (sub)gradient methods, with or without a diagonal preconditioner for adaptiveness, to induce a pre-specified type of desirable structure in the final model [49, 51, 9]. Unfortunately, although the added regularizer indeed induces some desirable structures at the stationary points of the training objective function, the iterates of these methods only converge to those stationary points asymptotically, but never really attain such a point at any iteration. Therefore, whether the output model of these algorithms, which is also an iterate that is only close enough to a stationary point, indeed possesses the ideal structure at the nearby stationary point is unknown, and theoretical analyses of these algorithms do not cover any guarantees regarding the obtained structure. Indeed, [16] oberserd empirically that the structures obtained by those methods are highly suboptimal and unstable over iterations. They then proposed a regularized dual averaging method called RMDA, and proved that after a finite number of steps, the iterates of RMDA can stably identify the locally optimal structure induced by the regularizer at the stationary point of asymptotic convergence.[1] This is up to our knowledge the only method with such structure guarantees for training structured neural networks. With this property, their experiments demonstrated that their method also empirically outperforms existing methods on modern computer vision tasks. However, since RMDA does not incorporate adaptiveness and their experiments are conducted only on medium-scale image classification problems, its usefulness beyond computer vision is in doubt.

For a wide range of tasks in deep learning such as language modeling and speech recognition, researchers have developed numerous architectures to achieve state-of-the-art prediction performance, including the transformer [44] and the LSTM [15]. The transformer is also gaining prominence in computer vision for achieving exceptional performance [29]. Therefore, it is becoming increasingly important to devise methods that attain satisfactory performance for training these network architectures with structure. For such modern architectures, adaptive methods like Adam [20] that iteratively rescale the stochastic gradient update directions via a coordinate-wise/diagonal preconditioner are known to outperform their non-adaptive counterparts and thus considered state-of-the-art [10, 1, 52, 28, 22]. It is hence expected that the non-adaptive RMDA of [16] might not lead to promising results for such widely-used architectures and tasks.

This work aims to fill this gap to propose a practical regularized adaptive method with guarantees for both convergence and structure identification. Since RMDA already has structure guarantees, it might look like we just need to combine it with an arbitrary preconditioner for adaptiveness. However, this seemingly easy extension actually requires deliberation in two aspects. First, except for few exceptions, combination of even a simple diagonal preconditioner and a nonsmooth regularizer makes the training subproblem complicated with no closed-form solution. This is totally different from adaptive methods with no regularization, whose subproblem optimal solution can be easily computed by coordinate-wise divisions. Therefore, in the regularized case, the best we can hope for is to apply an iterative approach to approximately solve the subproblem. This calls for careful design and control for the measure and the degree of the inexactness in the approximate subproblem solution. The second aspect is the need of an appropriate preconditioner that provides not only outstanding empirical performance but also desirable theoretical properties. The interplay between the inexactness and the preconditioner makes it particularly difficult to address the following three challenges simultaneously. (i) *Convergence*: Proving convergence of a new algorithm with even just one added component is always a nontrivial task. For example, although convergence of SGD has been well-studied for decades, similar guarantees for its adaptive correspondence, Adagrad, is not established until very recently [8]. We are dealing with both a preconditioner that changes the whole algorithm, just like from SGD to Adagrad, and the inevitable inexact subproblem solutions that could nullify many useful properties (regarding the subdifferential) commonly used in convergence proofs. (ii) *Structure*: Theoretical guarantees for structure identification is another critical aim of this work. Inexactness alone already makes this goal difficult; see Example 1 of [24] for a simple instance such that even

---

[1] See the first paragraph in Section 1 and Appendix B of [16] for a discussion about why the structure at the point of convergence is locally optimal.

infinitesimal inexactness could hinder structure identification. Even without inexactness, finding a preconditioner that leads to structure identification guarantees is already difficult because no adaptive algorithm, even in the much simpler deterministic and exact setting, is known to have such a guarantee. (iii) *Subproblem solver*: Our goal is a practical algorithm, so we need to solve the subproblem efficiently. This requires the inexact measure be checkable and the degree quickly attainable by a well-designed solver, and the preconditioner should make the subproblem well-conditioned and cannot complicate the computation of the solver.

To tackle these difficulties, we start from considering structure identification. We leverage the theory of manifold identification in variational analysis and nonlinear optimization to design a method that leads to finite-iteration structure identification guarantees. As discussed by [37, 16], the key to such guarantees for stochastic algorithms is to ensure the variance in the stochastic estimations decreases to zero. Due to the standard practice of data augmentation in deep learning, the training loss in the objective function is essentially the expected value of the training loss over a certain probability distribution instead of a finite sum. We thus draw inspirations from [25, 16] to consider a dual-averaging-type approach [32, 47] with momentum to attain variance reduction in this setting for the stochastic gradient estimation. However, we also need variance reduction for the preconditioner, so we carefully select a preconditioner whose update is in a manner similar to dual averaging, and prove that its variance also decreases to zero. We then conceive an implementable and practical subgradient-norm-based inexactness measure compatible with the structure identification theory. Further requirements are then added to the inexactness degree and the preconditioner to ensure convergence, and we also safeguard the preconditioner to keep the subproblems well-conditioned and the computation simple. We then propose to solve the subproblem by a proximal gradient (PG) solver that provably achieves our inexactness requirement efficiently. This leads to our Regularized Adaptive Momentum Dual Averaging (RAMDA) algorithm.

We summarize our main contributions as follows.

1. ***An adaptive algorithm for finding locally optimal structures***: RAMDA is the first regularized *adaptive* method guaranteed to find the locally optimal structure possessed by the stationary point to which its iterates converge. It thus produces models that are more structured while retaining the superb prediction performance of adaptive methods.

2. ***Efficient subproblem solver for regularized adaptive methods***: We propose an implementable inexactness condition and a companion efficient subproblem solver for regularized adaptive methods (including ours and existing ones) whose subproblems have no closed-form solution. We show that the induced inexactness does not affect convergence or structure identification guarantees. This condition and subproblem solver thus also serve as a key step for realizing existing frameworks for regularized adaptive methods.

3. ***A method with outstanding empirical performance***: Experiments on training modern neural networks in computer vision (ImageNet), language modeling (Transformer-XL), and speech (Tacotron2) with structured sparsity show that RAMDA steadily outperforms state of the art by achieving higher structured sparsity ratio and better prediction performance simultaneously.

## 2   Related Work

**Dual Averaging for Deep Learning.**   Our method is motivated by [16] that adapted the famous regularized dual averaging [47, 25] approach with momentum to train structured neural network models with data augmentation. They selected dual averaging for the gradient estimation to achieve variance reduction for structure guarantees, but their algorithm does not allow for adaptiveness. Inspired by this approach, we also take dual-averaging-like updates for the diagonal preconditioner in the subproblem for adaptiveness. Our preconditioner design also borrows ideas from the empirically successful MADGRAD of [7] for training non-regularized neural networks. RAMDA can thus also be seen as a generalization of MADGRAD to the regularized setting. Since no regularizer is present, unlike RAMDA, the subproblem of MADGRAD has a closed-form solution and no structure is expected. Moreover, [7] only analyzed convergence rates of the objective value when the problem is convex. Our analysis of (i) variance reduction in the preconditioner, (ii) convergence in the nonconvex nonsmooth regularized case, and (iii) structure identification guarantees are novel and closer

to properties desirable in practice. The first two items are also applicable when no regularizer is present, so our theory also expands guarantees for MADGRAD.

**Regularized Stochastic Algorithms for Deep Learning.** Other than RMDA, there are several works on training structured neural networks through regularization and its proximal operator, but none have structure guarantees. [49] considered a simple regularized SGD method with momentum, but their convergence analysis is only for the nonadaptive case. [51] studied a general regularized adaptive framework ProxGen that incorporates diagonal preconditioners, and showed that the subgradient of the objective function can decrease to the reciprocal of the batch size, but their result does not guarantee further convergence to stationary points. Moreover, they do not allow inexactness in the subproblem, so their framework can be realized for only a small class of problems. [9] proposed ProxSSI that extends ProxGen to the case of group-sparsity regularizers, whose corresponding subproblem indeed has no closed-form solution. They applied the Newton-Raphson method to obtain nearly-optimal subproblem solutions, and proposed a seemingly mild inexactness condition. Unfortunately, their condition is not checkable, and their corresponding convergence guarantee requires the regularizer to be locally smooth around each iterate, which excludes most regularizers that induce meaningful structures. On the other hand, we will show that with our implementable inexactness condition, ProxGen still possesses the same convergence guarantees in [51] without any additional requirement on the regularizer. Moreover, we will see in Section 6 that the time cost of the subproblem solver of ProxSSI is prohibitively high.

**Structure and Manifold Identification.** The major tool for our structure guarantees is the theory of manifold identification [12, 13, 27, 24] in variational analysis and nonlinear optimization. This theory shows that points possessing the same structure induced by the regularizer at a stationary point form a smooth manifold around this stationary point, and with properties from the regularizer, if a sequence of points converges to this stationary point with their corresponding subgradients decreasing to zero, this sequence is guaranteed to eventually stay in this manifold, thus identifying the structure. [25, 37, 16] have leveraged this tool to show manifold identification for various stochastic algorithms, and the common key, as pointed out by [37], is variance reduction. Our analysis uses a result given in [40] to prove so for both the gradient estimator and the preconditioner.

## 3 Problem Setting and Algorithm

As described in Section 1, we consider the case in which the training objective function is the expectation over a probability distribution as follows.

$$\min_{W \in \mathcal{E}} \quad F(W) := \mathbb{E}_{\xi \sim \mathcal{D}}[f_\xi(W)] + \psi(W), \tag{1}$$

where $\mathcal{E}$ is a Euclidean space with inner product $\langle \cdot, \cdot \rangle$ and its induced norm $\|\cdot\|$, $\mathcal{D}$ is a distribution over a space $\Omega$ representing all possible data modifications, $f_\xi$ is differentiable almost everywhere for any $\xi$, and the possibly nonsmooth regularizer $\psi(W)$ is for promoting a desirable structure in the optimal solutions.

Our algorithm can be seen as a double-dual averaging method that incorporates momentum, a proximal operation for the regularization, and dual averaging for updating both the stochastic gradient estimation and the preconditioner. For ease of description, we assume without loss of generality that $\mathcal{E} = \mathbb{R}^n$ in this section. At the $t$th iteration with learning rate $\eta_t$ and iterate $W^{t-1}$, we first draw an independent and identically distributed sample $\xi_t \sim \mathcal{D}$, compute the stochastic (sub)gradient $G^t := \nabla f_{\xi_t}(W^{t-1})$ of the loss function at the current point $W^{t-1}$ with respect to $\xi_t$, and then update the weighted sum $V_t$ of historical stochastic gradients and the weighted sum $U_t$ of their squared norms using the value $s_t$:

$$\begin{cases} V_0 := 0, & V_t := V_{t-1} + s_t G^t, & \forall t > 0, \\ U_0 := 0, & U_t := U_{t-1} + s_t G^t \circ G^t, & \forall t > 0, \end{cases} \quad s_t := \eta_t \sqrt{t}, \tag{2}$$

where $\circ$ denotes the Hadamard (pointwise) product in $\mathcal{E}$. We then construct the preconditioner $P^t$ and the weight sum $\alpha_t$ by

$$P^t := \text{Diag}(\sqrt[3]{U^t} + \epsilon), \quad \alpha_t := \sum\nolimits_{k=1}^{t} s_k, \tag{3}$$

**Algorithm 1** RAMDA $(W^0, T, T_2, \epsilon, \{\eta_t\}, \{c_t\}, \{\epsilon_t\})$

---

$V^0 \leftarrow 0, \quad U^0 \leftarrow 0, \quad \alpha_0 \leftarrow 0$
**for** $t = 1, \ldots, T$ **do**
$\quad$ Sample $\xi_t \sim \mathcal{D}, \quad s_t \leftarrow \eta_t \sqrt{t}, \quad \alpha_t \leftarrow \alpha_{t-1} + s_t, \quad G^t \leftarrow \nabla f_{\xi_t}(W^{t-1})$
$\quad$ Compute $V^t, U^t$ by (2) and construct $P^t$ by (3), and $\theta_t \leftarrow \max(\mathrm{diag}(P^t))^{-1}$
$\quad$ Compute $\hat{W}^t$ in (4) by PG $(W^{t-1}, W^0, \alpha_t^{-1} V^t, \alpha_t^{-1} P^t, \alpha_t \theta_t, T_2, \epsilon_t)$
$\quad$ Update $W^t$ by (5)
**output :** $W^T$

---

where $\epsilon > 0$ is a (usually small) constant for numerical stability and $\mathrm{Diag}(\cdot)$ is the diagonal matrix whose diagonal entries are the elements of the input vector. The update direction is then obtained by (approximately) solving the following subproblem.

$$\hat{W}^t \approx \arg\min_W \left( Q_t(W) \coloneqq \alpha_t \psi(W) + \langle V^t, W \rangle + \frac{1}{2} \langle W - W^0, P^t(W - W^0) \rangle \right), \quad (4)$$

where $W^0$ is the initial point. Details regarding (4) and how to solve it are deferred to Section 4. The iterate is then updated by averaging $\hat{W}^t$ and $W^{t-1}$ with some $c_t \in [0, 1]$:

$$W^t = (1 - c_t) W^{t-1} + c_t \hat{W}^t. \quad (5)$$

The choice of $P^t$ in (3) that uses the accumulated square of the stochastic gradient norm as the preconditioner is the key to adaptivity and is widely seen in adaptive methods such as Adagrad [11], while the choice of the cubic root instead of the square root is motivated by the impressive numerical performance of MADGRAD of [7] for smooth problems without a regularization term. The averaging step in (5) with $c_t \neq 1$ can be interpreted as incorporating a momentum term in the non-regularized non-adaptive case [43, 19].

## 4 Subproblem Solver

Given an iterate $W^{t-1}$, a momentum term $m_t$, a preconditioner $P^t$, and a stepsize $\eta_t$, existing regularized adaptive stochastic gradient algorithms for (1) can be summarized in the following form [51]:

$$W^t = \arg\min_W \left( \hat{Q}_t(W) \coloneqq \langle m_t, W \rangle + \frac{1}{2\eta_t} \langle W - W^{t-1}, P^t(W - W^{t-1}) \rangle + \psi(W) \right), \quad (6)$$

whose form is similar to (4). When the preconditioner $P^t$ is a multiple of the identity matrix like in the case of [16], the exact subproblem solution of (4) can be efficiently computed through the proximal operator associated with the regularizer. However, a major difficulty for realizing regularized adaptive methods, including the proposed RAMDA and the framework of [51] whose preconditioners are not a multiple of the identity, is that except for few special regularizers, the subproblem usually has no closed-form solution. We therefore consider using approximate solutions of the subproblem.

We propose to apply a few iterations of proximal gradient (PG) [see, *e.g.*, 5, 33] to approximately solve the subproblems in (4) and (6) when no closed-form solution is available, and we will show theoretically and empirically in the following sections that the inexactness of such approximate solutions has barely any effects on the theoretical guarantees and the final model quality. For the inexactness of the approximate solution in (4), we require

$$\min_{s \in \partial Q_t(\hat{W}^t)} \|s\| \leq \epsilon_t, \quad Q_t(\hat{W}^t) \leq Q_t(W^{t-1}), \quad (7)$$

for some pre-specified $\epsilon_t$, where $\partial Q_t(W^{t+1})$ is the (limiting) subdifferential [see, *e.g.*, 38, Definition 8.3]. This condition can be easily checked using information available in the PG iterations. For the sake of time efficiency, we also impose an upper limit for the number of PG iterations. Likewise, when applying our subproblem solver to (6), we enforce (7) but with $Q_t$ replaced by $\hat{Q}_t$ and $\hat{W}^t$ by $W^t$. We focus on the case of diagonal and positive $P^t$,

---

**Algorithm 2** PG($Z^0, W^0, V, P, \theta, T_2, \hat{\epsilon}$)

---

**if** $\psi$ *is nonconvex* **then** $\theta \leftarrow \theta/2$
**for** $j = 1, \ldots, T_2$ **do**
  $\quad Z^j \leftarrow \text{prox}_\psi(Z^{j-1} - \theta(V + P(Z^{j-1} - W^0)))$
  $\quad$**if** (7) *holds with* $\epsilon_t = \hat{\epsilon}$ *and* $\hat{W}^t = Z^j$ **then** $Z^{T_2} \leftarrow Z^j$, and break
**output**: $Z^{T_2}$

---

and thus the largest eigenvalue $\max(\text{diag}(P^t))$, where $\text{diag}(\cdot)$ is the vector formed by the diagonal entries of the input matrix, can be calculated easily and used to compute a step size guaranteeing sufficient objective decrease. For cases in which this value is difficult to obtain, one can apply a simple backtracking linesearch for the subproblem to find a suitable step size efficiently. This PG subproblem solver is summarized in Algorithm 2. To guarantee convergence for both our algorithm and the framework of [51], our analysis in Section 5 requires that $\{\epsilon_t\}$ satisfy

$$\bar{\epsilon} := \sum\nolimits_{t=0}^{\infty} \epsilon_t^2 < \infty. \tag{8}$$

We will show in Section 5 that (7) holds after at most $O(\epsilon_t^{-2})$ iterations of Algorithm 2.

## 5 Analysis

This section discusses theoretical guarantees for RAMDA and the proposed subproblem solver in Algorithm 2. We also prove convergence guarantees for applying PG to approximately solve (6) for the framework of [51]. All proofs are in the appendices. Some of our results are inspired by [16], but with the added inexactness in (4) and the adaptiveness for the preconditioner, the analysis is nontrivial. Recall that we assume that $f_\xi$ is differentiable only almost everywhere but not everywhere, which conforms with widely-used network structures like ReLU-type activations.

We first show that (7) can be attained by Algorithm 2 and that the point of convergence of RAMDA is almost surely a stationary point.

**Theorem 1.** *Assume that* (4) *and* (6) *has at least one optimal solution with a finite optimal objective value. Given* $\epsilon_t > 0$*, the number of iterations Algorithm* 2 *takes to satisfy* (7) *for both* (4) *and* (6) *is* $O(\log(\epsilon_t^{-1}))$ *when* $\psi$ *is convex and* $O(\epsilon_t^{-2})$ *when* $\psi$ *is nonconvex.*

**Theorem 2.** *Consider* $\{\hat{W}^t\}$ *generated by Algorithm 1 for* (1)*, with* (7) *and* $\{c_t\}$ *and* $\{\epsilon_t\}$ *satisfying* $\sum c_t = \infty$ *and* (8)*. Assume there is* $L \geq 0$ *such that for any* $\xi$*,* $f_\xi$ *is almost surely* $L$*-Lipschitz-continuously-differentiable, so the expectation is also* $L$*-Lipschitz-continuously-differentiable, there is* $C \geq 0$ *such that* $\mathbb{E}_{\xi_t \sim \mathcal{D}} \|\nabla f_{\xi_t}(W^{t-1})\|^4 \leq C$ *for all* $t$*, and that the set of stationary points* $\mathcal{Z} := \{W \mid 0 \in \partial F(W)\}$ *is nonempty. For any given* $W^0$*, consider the event that* $\{\hat{W}^t\}$ *converges to a point* $\bar{W}$ *(each event corresponds to a different* $\bar{W}$*). If* $\partial \psi$ *is outer semicontinuous at* $\bar{W}$*, this event has a nonzero probability, and* $\{\eta_t\}$ *satisfies*

$$\sum s_t \alpha_t^{-1} = \infty, \quad \sum \left(s_t \alpha_t^{-1}\right)^2 < \infty, \|W^{t+1} - W^t\| \left(s_t \alpha_t^{-1}\right)^{-1} \xrightarrow{a.s.} 0,$$

*then we have that* $\bar{W} \in \mathcal{Z}$ *with probability one conditional on this event. Moreover,* $\{W^t\}$ *also converges to this stationary point* $\bar{W}$*.*

Usually, convergence to a point requires some further regularity conditions like the Kurdyka–Łojasiewicz condition and boundedness of the iterates. However, existing frameworks regarding iterates convergence using such conditions also require the method analyzed to have a subgradient-descent-like behavior and to be a descent algorithm. Neither of these hold true even for the basic stochastic gradient algorithm, and we leave the analysis for this part as a challenging future work.

Our next key result shows that after a finite number of iterations, iterates of RAMDA all possess the same structure as that of the point of convergence $\bar{W}$. For this end, we first need to introduce the notions of partial smoothness and prox-regularity, and impose these assumptions on $\psi$ at $\bar{W}$.

**Definition 1** (Partial Smoothness [26, 12]). *A function $\psi$ is partly smooth at a point $\bar{W}$ relative to a set $\mathcal{M}_{\bar{W}} \ni \bar{W}$ if*

*1. Around $\bar{W}$, $\mathcal{M}_{\bar{W}}$ is a $\mathcal{C}^2$-manifold and $\psi|_{\mathcal{M}_{\bar{W}}}$ is $\mathcal{C}^2$.*

*2. $\psi$ is regular (finite and the Fréchet subdifferential coincides with the limiting Fréchet subdifferential) at all points $W \in \mathcal{M}_{\bar{W}}$ near $\bar{W}$, with $\partial\psi(W) \neq \emptyset$.*

*3. The affine span of $\partial\psi(\bar{W})$ is a translate of the normal space to $\mathcal{M}_{\bar{W}}$ at $\bar{W}$.*

*4. $\partial\psi$ is continuous at $\bar{W}$ relative to $\mathcal{M}_{\bar{W}}$.*

We often call $\mathcal{M}_{\bar{W}}$ the active manifold at $\bar{W}$. Locally, this manifold represents all points near $\bar{W}$ that share the same structure induced by the regularized as $\bar{W}$. Therefore, finding the active manifold is equivalent to finding the locally optimal structure.

**Definition 2** (Prox-regularity [36]). *A function $\psi$ is prox-regular at $\bar{W}$ for $V^* \in \partial\psi(\bar{W})$ if $\psi$ is locally lower semi-continuous around $\bar{W}$, finite at $\bar{W}$, and there is $\rho > 0$ such that $\psi(W_1) \geq \psi(W_2) + \langle V, W_1 - W_2 \rangle - \frac{\rho}{2}\|W_1 - W_2\|^2$ for every $W_1, W_2$ near $\bar{W}$ with $\psi(W_2)$ close to $\psi(\bar{W})$ and $V \in \partial\psi(W_2)$ close to $V^*$. $\psi$ is prox-regular at $\bar{W}$ if it is prox-regular for all $V \in \partial\psi(\bar{W})$.*

**Theorem 3.** *Consider Algorithm 1 with the conditions in Theorem 2 satisfied. Consider the event of $\{\hat{W}^t\}$ converging to a certain point $\bar{W}$ as in Theorem 2. If the probability of this event is nonzero; $\psi$ is prox-regular and subdifferentially continuous at $\bar{W}$ and partly smooth at $\bar{W}$ relative to the active $\mathcal{C}^2$ manifold $\mathcal{M}_{\bar{W}}$; $\partial\psi$ is outer semicontinuous at $\bar{W}$; and the nondegeneracy condition $-\nabla f(\bar{W}) \in$ relative interior of $\partial\psi(\bar{W})$ holds at $\bar{W}$, then conditional on this event, almost surely there is $T_0 \geq 0$ such that*

$$\hat{W}^t \in \mathcal{M}_{\bar{W}}, \quad \forall t \geq T_0.$$

We note particularly that convex and weakly-convex [34] functions are all regular, prox-regular, and subdifferentially continuous everywhere.

We also show that our subproblem solver and condition can be effectively applied to the framework of [51] while retaining the same convergence guarantees. As mentioned in Section 2, our result is much stronger than that of [9] for having no unrealistic smoothness requirement on $\psi$ and using an implementable inexactness condition.

**Theorem 4.** *For the framework in [51] with the subproblem solved approximately by Algorithm 2 such that (7) holds with $\{\epsilon_t\}$ satisfying (8). Then Theorem 1 of [51] still holds, but with the constants $\{Q_i\}$ being also dependent on $\bar{\epsilon}$.*

## 6 Experiments

This section examines the practical performance of RAMDA for training structured neural networks. As sparsity is arguably one of the most widely adopted structures in machine learning, we follow [45] to consider structured sparsity as the representative structure in our experiments. Particularly, we employ the group LASSO regularization [50] to encourage group sparsity and disable weight decay in all experiments, except for the dense baselines. We begin from examining the efficiency and effectiveness of PG for both RAMDA and existing regularized adaptive methods. We then consider tasks in computer vision, language modeling, and speech to compare the following algorithms using Pytorch.

- RAMDA: The proposed Algorithm 1.
- RMDA [16]
- ProxSGD [49]
- ProxGen [51]: We follow their experiments to use AdamW and apply our PG as the subproblem solver.
- ProxSSI [9]

These algorithms are introduced in Section 2 and also further summarized in Appendix A. For each task, we also provide for reference a baseline that does not include a group LASSO regularizer in the training (SGD with momentum (MSGD) for computer vision, and Adam

Table 1: Weighted group sparsity and validation accuracy of different subproblem stopping criteria.

| Model/Data | Algorithm | No early stopping | | Early stopping | |
|---|---|---|---|---|---|
| | | Accuracy | Sparsity | Accuracy | Sparsity |
| VGG19 / | ProxGen | $92.7 \pm 0.2\%$ | $88.8 \pm 0.0\%$ | $92.7 \pm 0.1\%$ | $86.9 \pm 0.4\%$ |
| CIFAR10 | RAMDA | $92.7 \pm 0.2\%$ | $86.7 \pm 0.3\%$ | $92.9 \pm 0.2\%$ | $86.3 \pm 0.4\%$ |
| ResNet50 / | ProxGen | $73.6 \pm 0.1\%$ | $74.7 \pm 0.6\%$ | $74.0 \pm 0.1\%$ | $67.6 \pm 3.1\%$ |
| CIFAR100 | RAMDA | $69.9 \pm 1.5\%$ | $69.5 \pm 2.1\%$ | $71.2 \pm 1.4\%$ | $67.5 \pm 1.6\%$ |

for the other two), but our comparison is only among those for training structured models. Our code for reproducing the experiments and the hyperparameter settings are available at `https://github.com/ismoptgroup/ramda_exp/`. Additional details of the stability of the structure (level of structured sparsity here) over epochs of RAMDA are available in Appendix D.

We use two criteria for comparison: 1. Model predictive ability, and 2. Structured sparsity level. The former is task-dependent and thus specified in each experiment. Regarding the latter, sparsifying neural networks while preserving its performance requires prior knowledge of model design. A common approach is retaining certain parameters during the training process, and we adhere to this convention such that the bias, batch normalization [17], layer normalization [3], and embedding layers do not have any sparsity-inducing regularization imposed on them [9, 35]. For the rest, we adopt channel-wise grouping for convolutional layers, input-wise grouping for fully-connected and LSTM layers during the training phase. For evaluation, our structured sparsity is calculated using the weighted group sparsity with the weights proportional to the number of parameters in each group.

We run each experiment with three different random initializations and show the mean and standard deviation of the validation predictive performance and the structured sparsity of the final model of all methods.

**Subproblem**   We start from showing the effectiveness of our proposed subproblem solver for RAMDA and ProxGen. For both approaches, we use Theorem 2 of [9] to safely screen out a portion of groups that will be zero at the optimal subproblem solution, and opt for the PG algorithm to solve the remaining parts. We consider two practical stopping criteria for PG: 1. Running until it reaches the maximum iterations (no early stopping), and 2. Terminate when the subproblem objective improvement is small (early stopping). For the former, we set the maximum to 100. For the latter, we terminate PG early if $(Q_t(Z^{j-1}) - Q_t(Z^j))/(|Q_t(Z^j)| + 1) < 10^{-8}$ is reached. Moreover, to ensure incorporation of the preconditioner into ProxGen, we set its minimum PG iterations to 2. We examine how these stopping criteria affect the final model of RAMDA and ProxGen using image classification problems of a smaller scale. From Table 1, we see that early stopping does not affect the outcome much. Given that early stopping is more efficient, we will adopt it in all subsequent experiments.

Next, we compare ProxGen with ProxSSI (these two only differ in the subproblem solver) to examine the efficiency and performance differences between solving the subproblems approximately and (almost) exactly in Table 2. We see that our solver is around 3X faster than ProxSSI, and the model qualities are similar. We thus exclude ProxSSI from our comparisons in the following experiments due to its excessively lengthy running time, especially for large-scale tasks.

**Image Classification**   We conduct a classical computer vision task of training ResNet50 [14] with the ILSVRC 2012 ImageNet dataset [39]. The result in Table 3 shows that RAMDA attains the best validation accuracy and structured sparsity simultaneously.

**Language Modeling**   For language modeling, we train Transformer-XL (base) [6] using the WikiText-103 dataset [31]. Transformer-XL is comprised of embedding and non-embedding layers, and in the PyTorch implementation, the non-embedding layers are built using linear

Table 2: Weighted group sparsity, validation accuracy and time/epoch of ProxSSI and ProxGen for CIFAR10/CIFAR100. We report the average time/epoch using one NVIDIA V100 GPU.

| Algorithm | Accuracy | Sparsity | Time | Accuracy | Sparsity | Time |
|---|---|---|---|---|---|---|
| | VGG19/CIFAR10 | | | VGG19/CIFAR100 | | |
| ProxSSI | $92.8 \pm 0.1\%$ | $88.4 \pm 0.2\%$ | 79s | $67.3 \pm 0.1\%$ | $78.6 \pm 0.3\%$ | 79s |
| ProxGen | $92.8 \pm 0.0\%$ | $86.6 \pm 0.1\%$ | 24s | $68.1 \pm 0.4\%$ | $75.5 \pm 0.2\%$ | 26s |
| | ResNet50/CIFAR10 | | | ResNet50/CIFAR100 | | |
| ProxSSI | $94.0 \pm 0.1\%$ | $83.7 \pm 0.6\%$ | 260s | $73.7 \pm 0.4\%$ | $70.4 \pm 0.7\%$ | 251s |
| ProxGen | $94.1 \pm 0.1\%$ | $80.4 \pm 0.4\%$ | 70s | $73.6 \pm 0.4\%$ | $65.5 \pm 3.6\%$ | 74s |

Table 3: Weighted group sparsity and validation accuracy on ImageNet/ResNet50.

| Algorithm | Accuracy | Sparsity |
|---|---|---|
| MSGD | $77.14 \pm 0.04\%$ | - |
| RAMDA | $\mathbf{74.53 \pm 0.10\%}$ | $\mathbf{29.19 \pm 0.94\%}$ |
| ProxSGD | $73.50 \pm 0.20\%$ | $17.54 \pm 1.26\%$ |
| ProxGen | $74.17 \pm 0.08\%$ | $20.29 \pm 0.22\%$ |
| RMDA | $74.47 \pm 0.08\%$ | $25.20 \pm 1.69\%$ |

and layer-normalization layers. We apply group LASSO regularization to the linear layers, and present in Table 4 the perplexity and the weighted group sparsity of the models trained. We see that RAMDA gives the best perplexity and structured sparsity simultaneously.

**Speech Synthesis**  We consider Tacotron2 [41] for speech synthesis on the LJSpeech dataset [18]. We apply regularization to the convolutional, LSTM, and linear layers of Tacotron2 and show the results in Table 5. Clearly, RAMDA gives the lowest validation loss and the highest group sparsity.

**Time Efficiency**  In Tables 4 and 5, we see that although RAMDA and ProxGen have more difficult subproblems without a closed-form solution to solve, our proposed PG solver is highly efficient such that the running time of them is still close to other approaches, making these regularized adaptive approaches practically feasible.

**Summary**  In summary, thanks to its adaptive nature (for better predictive performance) and its ability of manifold identification (for higher structured sparsity), RAMDA is superior to state of the art on modern language modeling and speech synthesis tasks as well as the ImageNet problem. We also observe from the plots in the appendices that it is possible to further improve the sparsity level of RAMDA if we run it for more epochs.

## 7   Conclusions

In this work, we proposed a regularized dual averaging method with adaptiveness, RAMDA, for training structured neural networks. Our method outperforms state of the art on modern

Table 4: Weighted group sparsity and validation perplexity on Transformer-XL with WikiText-103.

| Alg. | Perplexity | Sparsity | Time/epoch |
|---|---|---|---|
| Adam | $23.00 \pm 0.05$ | - | $6261 \pm 21$s |
| RAMDA | $\mathbf{26.97 \pm 0.10}$ | $\mathbf{36.2 \pm 0.3\%}$ | $6954 \pm 30$s |
| ProxSGD | $27.42 \pm 0.02$ | $33.1 \pm 1.5\%$ | $6167 \pm 12$s |
| ProxGen | $27.49 \pm 0.19$ | $30.5 \pm 0.6\%$ | $6652 \pm 21$s |
| RMDA | $27.10 \pm 0.08$ | $36.0 \pm 2.7\%$ | $6184 \pm 20$s |

Table 5: Weighted group sparsity and validation loss on Tacotron2 with LJSpeech.

| Alg. | Loss | Sparsity | Time/epoch |
|------|------|----------|------------|
| Adam | $0.39 \pm 0.02$ | - | $431 \pm 2s$ |
| RAMDA | $\mathbf{0.44 \pm 0.01}$ | $\mathbf{52.9 \pm 1.6\%}$ | $443 \pm 1s$ |
| ProxSGD | $0.50 \pm 0.00$ | $34.3 \pm 1.6\%$ | $431 \pm 0s$ |
| ProxGen | $0.45 \pm 0.01$ | $45.6 \pm 0.9\%$ | $438 \pm 2s$ |
| RMDA | $0.46 \pm 0.01$ | $45.9 \pm 1.7\%$ | $431 \pm 2s$ |

architectures including LSTM and transformers as well as the ImageNet problem. We also proposed a subroutine with strong convergence guarantees to approximately solve the regularized subproblem of both our method and an existing framework efficiently. Extensive experiments on group sparsity showed that our subproblem solver can greatly reduce the training time for existing methods, and our proposed RAMDA achieves simultaneously higher structured sparsity ratio and better prediction performance than existing methods. Implementation of our method is available at `https://www.github.com/zhisyuan1214/RAMDA/`.

## Acknowledgement

Ching-pei's research is supported in part by the JSPS Grant-in-Aid for Research Activity Start-up 23K19981 and Grant-in-Aid for Early-Career Scientists 24K20845.

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

# Appendices

## Table of Contents

## A  More Experiment Details

This section describes details of our implementation of RAMDA and the setting of the experiments conducted in Section 6.

## A.1 Implementation and Hyperparameter Setting of RAMDA

Similar to [16], we introduce a restart strategy to the implementation of RAMDA. During each stage of the training, the learning rate $\eta_t$ and the momentum factor $c_t$ are fixed. Once the epoch count enters the next stage, we reset the counter $t$ to 1 and use the output parameter $W^T$ from the previous round as the new input parameter $W^0$ to the same algorithm, set $\alpha_t, V^t$ and $U^t$ to 0, but keep the scheduling for $\eta$ and $c$ going without resetting them, and decrease $\epsilon$ by a factor. We initialize $c_t$ as either 0.1 or 0.01, depending on the problems, and use a constant $c_t$ until the final stage, where we gradually increase it by

$$c_t = \min(c\sqrt{i}, 1),$$

where $i$ counts the training steps executed at this final stage. This momentum strategy is applied to both RAMDA and RMDA in our experiments.

## A.2 Details of the Algorithms Compared

In Table 6, we summarize details of the algorithms compared in Section 6.

Table 6: Algorithms used in the experiments.

| Algorithm | Unregularized counterpart | Subproblem |
|---|---|---|
| RAMDA | MADGRAD [7] | PG |
| RMDA | MDA [19] | Closed-form solution |
| ProxSGD | MSGD | Closed-form solution |
| ProxGen | AdamW [30] | PG |
| ProxSSI | AdamW [30] | Newton-Raphson |

## A.3 Computational Resource

We conduct all experiments utilizing NVIDIA TESLA V100 (32 GB) GPUs. We employ eight V100 GPUs for each run of the ImageNet experiments. For all other experiments, we utilize a single V100 GPU per run.

# B Proofs

This section provides proofs of the theoretical results stated in Section 5. We restate these results and provide their corresponding proofs right after each statement.

**Theorem 1.** *Assume that* (4) *and* (6) *has at least one optimal solution with a finite optimal objective value. Given $\epsilon_t > 0$, the number of iterations of Algorithm 2 takes to satisfy* (7) *for both* (4) *and* (6) *is $O(\epsilon_t^{-1})$ when $\psi$ is convex and $O(\epsilon_t^{-2})$ when $\psi$ is nonconvex.*

*Proof.* We use the notation

$$\bar{Q}_t(Z) = f_t(Z) + \psi(Z)$$

to unify the two objective function $Q_t/\alpha_t$ and $\hat{Q}_t$, where $f_t$ is the smooth part and we define the Lipschitz constant of $\nabla f_t$ as $L$.

At each iteration, PG solves the following subproblem

$$Z^{j+1} \in \arg\min_Z \quad \langle \nabla f_t(Z^j), Z - Z^j \rangle + \frac{1}{2\theta_t} \big\| Z - Z^j \big\|^2 + \psi(Z),$$

and thus from the first-order optimality conditions, clearly we have

$$\nabla f_t(Z^{j+1}) - \nabla f_t(Z^j) - \frac{1}{\theta_t} \big( Z^{j+1} - Z^j \big) \in \partial \bar{Q}_t(Z^{j+1}).$$

We thus have from the Lipschitz continuity of $\nabla f_t$ that

$$\min_{s \in \partial \bar{Q}_t(Z^{j+1})} \|s\| \le \big\| \nabla f_t(Z^{j+1}) - \nabla f_t(Z^j) \big\| + \frac{1}{\theta_t} \big\| Z^{j+1} - Z^j \big\| \le \big( L + \theta_t^{-1} \big) \big\| Z^{j+1} - Z^j \big\|.$$

$$(9)$$

Note that $\bar{Q}_t$ is lower bounded, say by $\bar{Q}_t^*$, and has at least one solution $Z^*$ (unique when $\psi$ is convex).

In the case that $\psi$ is convex, we know that $\theta_t = 1/L$, and (3) clearly shows that the subproblem objective $\bar{Q}$ is $\epsilon$-strongly convex. Therefore, standard analysis of proximal gradient [see, for example, 4, Lemma 10.4 and iTheorem 10.29] gives that

$$\frac{L}{2}\left\|Z^{j+1} - Z^j\right\| \leq \bar{Q}_t(Z^j) - \bar{Q}_t(Z^{j+1}), \quad \forall j \geq 0, \tag{10}$$

$$\bar{Q}_t(Z^j) - \bar{Q}_t(Z^*) \leq \frac{L}{2}\left(1 - \frac{\epsilon}{L}\right)^j \left\|Z^0 - Z^*\right\|^2, \quad \forall j \geq 1. \tag{11}$$

The combination of (9)–(11) shows that it takes $O(\log \epsilon_t^{-1})$ iterations for PG to reach the required precision of $\epsilon_t$.

When $\psi$ is nonconvex, we have that $\theta_t = 1/(2L)$ and standard analysis [2, Section 5.1] gives

$$\min_{k=0,1,\ldots,j}\left\|Z^{j+1} - Z^j\right\| \leq \frac{C}{\sqrt{j}} \tag{12}$$

for some constant $C$ depending on $L$ and $\bar{Q}_t(W^t) - \bar{Q}_t^*$. Therefore, (12) and (9) show that it takes $O(\epsilon_t^{-2})$ iterations to reach the desired precision. $\qquad\square$

**Theorem 2.** *Consider $\{\hat{W}^t\}$ generated by Algorithm 1 for (1), with (7) and $\{c_t\}$ and $\{\epsilon_t\}$ satisfying $\sum c_t = \infty$ and (8). Assume there is $L \geq 0$ such that for any $\xi$, $f_\xi$ is almost surely $L$-Lipschitz-continuously-differentiable, so the expectation is also $L$-Lipschitz-continuously-differentiable, there is $C \geq 0$ such that $\mathbb{E}_{\xi_t \sim \mathcal{D}}\left\|\nabla f_{\xi_t}\left(W^{t-1}\right)\right\|^4 \leq C$ for all $t$, and that the set of stationary points $\mathcal{Z} := \{W \mid 0 \in \partial F(W)\}$ is nonempty. For any given $W^0$, consider the event that $\{\hat{W}^t\}$ converges to a point $\bar{W}$ (each event corresponds to a different $\bar{W}$). If $\partial\psi$ is outer semicontinuous at $\bar{W}$, this event has a nonzero probability, and $\{\eta_t\}$ satisfies*

$$\sum s_t \alpha_t^{-1} = \infty, \quad \sum\left(s_t \alpha_t^{-1}\right)^2 < \infty, \tag{13}$$

$$\left\|W^{t+1} - W^t\right\|\left(s_t \alpha_t^{-1}\right)^{-1} \xrightarrow{a.s.} 0, \tag{14}$$

*then we have that $\bar{W} \in \mathcal{Z}$ with probability one conditional on this event. Moreover, $\{W^t\}$ also converges to this stationary point $\bar{W}$.*

*Proof.* First, we prove that when $\{\hat{W}^t\}$ converges to $\bar{W}$, $\{W^t\}$ also converges to $\bar{W}$. From (5), we have that

$$\left\|W^t - \bar{W}\right\| \leq (1 - c_t)\left\|W^{t-1} - \bar{W}\right\| + c_t\left\|\hat{W}^t - \bar{W}\right\|. \tag{15}$$

Since $\hat{W}^t \to \bar{W}$, for any $\epsilon > 0$ we can find an integer $t_\epsilon \geq 0$ such that $\left\|\hat{W}^t - \bar{W}\right\| \leq \epsilon$ for all $t \geq t_\epsilon$. Therefore, by deducting $\epsilon$ from both sides of (15), we get

$$\left\|W^t - \bar{W}\right\| - \epsilon \leq \left(\prod_{k=t_\epsilon}^t (1 - c_t)\right)\left(\left\|W^{t_\epsilon - 1} - \bar{W}\right\| - \epsilon\right) \leq \exp\left(-\sum_{k=t_\epsilon}^t c_t\right)\left(\left\|W^{t_\epsilon - 1} - \bar{W}\right\| - \epsilon\right), \quad \forall t \geq t_\epsilon.$$

By letting $t$ approach infinity and noting that $\sum c_t = \infty$, we see that

$$\lim_{t \to \infty}\left\|W^t - \bar{W}\right\| \leq \epsilon.$$

Because $\epsilon$ is arbitrary, we see that $\left\|W^t - \bar{W}\right\| \to 0$, and hence $\{W^t\}$ converges to $\bar{W}$.

Next, consider the update of $\alpha_t^{-1} U^t$, we can see from (2) that

$$\frac{U^t}{\alpha_t} = \frac{\alpha_{t-1}}{\alpha_t}\frac{U^{t-1}}{\alpha_{t-1}} + \frac{s_t \nabla f_{\xi_t}(W^{t-1})}{\alpha_t} = \left(1 - \frac{s_t}{\alpha_t}\right)\frac{U^{t-1}}{\alpha_{t-1}} + \frac{s_t}{\alpha_t}\nabla f_{\xi_t}(W^{t-1}). \tag{16}$$

Moreover, the assumptions on $\eta_t$ satisfies all the required conditions of Lemma 1 of [40]. We therefore apply Lemma 1 of [40] to conclude that

$$\frac{U^t}{\alpha_t} \xrightarrow{\text{a.s.}} \mathbb{E}_{\xi \sim \mathcal{D}} \left[ \nabla f_\xi \left( W^t \right) \circ \nabla f_\xi \left( W^t \right) \right].  \tag{17}$$

The update for $\alpha_t^{-1} V^t$ has a form analogous to (16), and we have from Jensen's inequality that

$$\mathbb{E}_{\xi_t \sim \mathcal{D}} \left\| \nabla f_{\xi_t} \left( W^{t-1} \right) \right\|^2 \leq \sqrt{\mathbb{E}_{\xi_t \sim \mathcal{D}} \| \nabla f_{\xi_t} \left( W^{t-1} \right) \|^4} \leq \sqrt{C},$$

implying that the second moment is also bounded in expectation. We can therefore also apply Lemma 1 of [40] to $\alpha_t^{-1} V^t$ and conclude that

$$\frac{V^t}{\alpha_t} \xrightarrow{\text{a.s.}} \nabla \mathbb{E}_{\xi \sim \mathcal{D}} \left[ f_\xi \left( W^t \right) \right].  \tag{18}$$

We further notice that the union of two events that happens almost surely is still an event that happens almost surely.

From (4) and (7), we can see that there is a sequence $\{z_t\}$ such that

$$-\left( \frac{V^t}{\alpha_t} + \frac{z_t}{\alpha_t} + \frac{P^t}{\alpha_t} (\hat{W}^t - W^0) \right) \in \partial \psi(\hat{W}^t), \quad \|z_t\| \leq \epsilon_t.  \tag{19}$$

Our assumption in (14) implies that $\alpha_t \to \infty$, which together with (8) leads to

$$\frac{z_t}{\alpha_t} \to 0.  \tag{20}$$

From (18), that $\nabla \mathbb{E}_{\xi \sim \mathcal{D}} \left[ f_\xi \left( W \right) \right]$ is Lipschitz continuous, and that $W^t \to \bar{W}$ (which we have proven in the first part), we see that

$$\frac{V^t}{\alpha_t} \xrightarrow{\text{a.s.}} \nabla \mathbb{E}_{\xi \sim \mathcal{D}} \left[ f_\xi(\bar{W}) \right].  \tag{21}$$

For the third term, we have from (3) and (17) that

$$\frac{P^t}{\alpha_t} = \alpha_t^{-\frac{2}{3}} \text{Diag} \left( \sqrt[3]{\frac{U^t}{\alpha_t}} \right) + \frac{\epsilon}{\alpha_t} I.$$

Again since $\alpha_t \to \infty$, the second term of the equation above converges to 0. Therefore, by (17), we obtain that

$$\frac{P^t}{\alpha_t} \xrightarrow{\text{a.s.}} \alpha_t^{-\frac{2}{3}} \text{Diag} \left( \sqrt[3]{\mathbb{E}_{\xi \sim \mathcal{D}} \left[ \nabla f_\xi \left( W^t \right) \circ \nabla f_\xi \left( W^t \right) \right]} \right).$$

Again from the continuity of $\nabla \mathbb{E}_{\xi \sim \mathcal{D}} \left[ f_\xi \left( W \right) \right]$ and that $\alpha_t \to \infty$, we conclude that

$$\frac{P^t}{\alpha_t} \xrightarrow{\text{a.s.}} \alpha_t^{-\frac{2}{3}} \text{Diag} \left( \sqrt[3]{\mathbb{E}_{\xi \sim \mathcal{D}} \left[ \nabla f_\xi \left( \bar{W} \right) \circ \nabla f_\xi \left( \bar{W} \right) \right]} \right) \xrightarrow{\text{a.s.}} 0.  \tag{22}$$

Finally, using the outer semicontinuity of $\partial \psi(W)$ at $\bar{W}$, we conclude from (19)–(22) that

$$0 \in \nabla \mathbb{E}_{\xi \sim \mathcal{D}} \left[ f_\xi \left( \bar{W} \right) \right] + \lim_{t \to \infty} \psi(\hat{W}^t) \subseteq \nabla \mathbb{E}_{\xi \sim \mathcal{D}} \left[ f_\xi \left( \bar{W} \right) \right] + \psi(\bar{W}) = \partial F(\bar{W})$$

with probability one, showing that $\bar{W}$ is a stationary point almost surely. $\qquad \square$

**Theorem 3.** *Consider Algorithm 1 with the conditions in Theorem 2 satisfied. Consider the event of $\{\hat{W}^t\}$ converging to a certain point $\bar{W}$ as in Theorem 2. If the probability of this event is nonzero; $\psi$ is prox-regular and subdifferentially continuous at $\bar{W}$ and partly smooth at $\bar{W}$ relative to the active $\mathcal{C}^2$ manifold $\mathcal{M}_{\bar{W}}$; $\partial \psi$ is outer semicontinuous at $\bar{W}$; and the nondegeneracy condition*

$$- \nabla f \left( \bar{W} \right) \in \text{relative interior of } \partial \psi \left( \bar{W} \right)  \tag{23}$$

*holds at $\bar{W}$, then conditional on this event, almost surely there is $T_0 \geq 0$ such that*

$$\hat{W}^t \in \mathcal{M}_{\bar{W}}, \quad \forall t \geq T_0.  \tag{24}$$

*In other words, the active manifold at $\bar{W}$ is identified by the iterates of Algorithm 1 after a finite number of iterations almost surely.*

*Proof.* From (19), there exists a sequence $\{Y^t\}$ such that

$$Y^t \in \partial\psi(\hat{W}^t), \quad \frac{V^t}{\alpha_t} + \frac{z_t}{\alpha_t} + \frac{P^t}{\alpha_t}(\hat{W}^t - W^0) + Y^t = 0, \quad \forall t. \tag{25}$$

For notational ease, we denote

$$f(W) := \mathbb{E}_{\xi \sim \mathcal{D}}[f_\xi(W)]. \tag{26}$$

From (25), we then get

$$\nabla f(\hat{W}^t) - \frac{V^t}{\alpha_t} - \frac{z_t}{\alpha_t} - \frac{P^t}{\alpha_t}(\hat{W}^t - W^0) \in \partial F(\hat{W}^t). \tag{27}$$

We aim to show that

$$\text{dist}(0, \partial F(\hat{W}^t)) := \min_{Y \in \partial F(\hat{W}^t)} \|Y\|$$

converges to 0 almost surely. From (27), we have

$$\begin{aligned}
\text{dist}(0, \partial F(\hat{W}^t)) &\leq \left\| \nabla f(\hat{W}^t) - \frac{V^t}{\alpha_t} - \frac{z_t}{\alpha_t} - \frac{P^t}{\alpha_t}(\hat{W}^t - W^0) \right\| \\
&\leq \left\| \nabla f(\hat{W}^t) - \frac{V^t}{\alpha_t} \right\| + \left\| \frac{z_t}{\alpha_t} \right\| + \left\| \frac{P^t}{\alpha_t}(\hat{W}^t - W^0) \right\| \\
&\leq \left\| \nabla f(\hat{W}^t) - \frac{V^t}{\alpha_t} \right\| + \frac{\epsilon_t}{\alpha_t} + \left\| \frac{P^t}{\alpha_t}(\hat{W}^t - W^0) \right\|,
\end{aligned} \tag{28}$$

where we get the first inequality from the triangle inequality and the second from (19). According to (18) and (22), there are $\{A_t\}$ and $\{B_t\}$ such that

$$\begin{cases} \frac{V^t}{\alpha_t} = \nabla f(W^t) + A_t, \quad \|A_t\| \xrightarrow{\text{a.s.}} 0 \\ \frac{P^t}{\alpha_t} = \alpha_t^{-\frac{2}{3}} \text{Diag}\left( \sqrt[3]{\nabla f(W^t) \circ \nabla f(W^t)} \right) + B_t, \quad \|B_t\| \xrightarrow{\text{a.s.}} 0. \end{cases} \tag{29}$$

Substituting the above two equations back to (28), we obtain

$$\begin{aligned}
&\text{dist}(0, \partial F(\hat{W}^t)) \\
&\leq \left\| \nabla f(\hat{W}^t) - \nabla f(W^t) \right\| + \|A_t\| + \frac{\epsilon_t}{\alpha_t} + \left( \alpha_t^{-\frac{2}{3}} \left\| \sqrt[3]{\nabla f(W^t) \circ \nabla f(W^t)} \right\|_\infty + \|B_t\| \right) \left\| \hat{W}^t - W^0 \right\| \\
&\leq L \left\| \hat{W}^t - W^t \right\| + \|A_t\| + \frac{\epsilon_t}{\alpha_t} + \left( \alpha_t^{-\frac{2}{3}} \left\| \sqrt[3]{\nabla f(W^t) \circ \nabla f(W^t)} \right\|_\infty + \|B_t\| \right) \left\| \hat{W}^t - W^0 \right\|.
\end{aligned} \tag{30}$$

From Theorem 2, we know that $\hat{W}^t$ and $W^t$ both converge to $\bar{W}$, so

$$\left\| \hat{W}^t - W^t \right\| \leq \left\| \hat{W}^t - \bar{W} \right\| + \left\| W^t - \bar{W} \right\| \to 0.$$

From (8) and (14), we know that $\epsilon_t/\alpha_t \to 0$. Because $\hat{W}^t \to \bar{W}$, we also have that

$$\left\| \hat{W}^t - W^0 \right\| \to \left\| \bar{W} - W^0 \right\| < \infty.$$

From $W^t \to \bar{W}$, we have that

$$\left\| \sqrt[3]{\nabla f(W^t) \circ \nabla f(W^t)} \right\|_\infty \to \left\| \sqrt[3]{\nabla f(\bar{W}) \circ \nabla f(\bar{W})} \right\|_\infty < \infty.$$

Combining these results with (30), we conclude that

$$L \left\| \hat{W}^t - W^t \right\| + \|A_t\| + \frac{\epsilon_t}{\alpha_t} + \left( \alpha_t^{-\frac{2}{3}} \left\| \sqrt[3]{\nabla f(W^t) \circ \nabla f(W^t)} \right\|_\infty + \|B_t\| \right) \left\| \hat{W}^t - W^0 \right\| \xrightarrow{\text{a.s.}} 0,$$

proving that

$$\text{dist}(0, \partial F(\hat{W}^t)) \xrightarrow{\text{a.s.}} 0.$$

On the other hand, since $f$ is continuous and $\psi$ is subdifferentially continuous at $\bar{W}$ (which implies $F$ is also subdifferentially contnuous at $\bar{W}$), $\hat{W}^t \to \bar{W}$, and that $\nabla f(\hat{W}^t) + Y_t \xrightarrow{\text{a.s.}} 0 \in \partial F(\bar{W})$ (from Theorem 2), we know that $F(\hat{W}^t) \xrightarrow{\text{a.s.}} F(\bar{W})$ as well. Therefore, we can apply Lemma 1 of [24] to conclude that (24) indeed holds for some $T_0 < \infty$ almost surely. $\quad\square$

**Algorithm 3** ProxGen $(W^0, T, T_2, \{\eta_t\}, \{\rho_t\}, \{c_t\}, \{\epsilon_t\}, \{b_t\}, \delta I)$

---

$m_0 \leftarrow 0$
**for** $t = 1, \dots, T$ **do**
  Sample $\xi_t \sim \mathcal{D}$ with batch size $b_t$
  $G^t \leftarrow \nabla f_{\xi_t}(W^{t-1})$
  $m_t \leftarrow \rho_t m_{t-1} + (1 - \rho_t)G^t$
  Construct $P^t$ satisfying $P^t \succeq \delta I$
  $\theta_t \leftarrow 1/\|P^t\|_2$
  Compute $W^{t+1}$ by roughly solving (6) that satisfies (7) with $Q_t$ replaced by $\hat{Q}_t$ and $\hat{W}^t$
  replaced by $W^{t+1}$, using PG $(W^t, W^t, m_t, \eta_t^{-1}P^t, \theta_t, T_2, \epsilon_t)$
**output :** $W^T$

---

### B.1 Convergence Result for ProxGen

We next discuss the convergence result for the framework of [51] with inexactness added. For consistency, we first use our notations to introduce their framework, with our inexactness condition added, in Algorithm 3.

In their analysis, [51] made the following four assumptions, and we will follow these assumptions using the notation (26).

**(C-1)** The expected loss function $f$ is $L$-Lipschitz-continuously-differentiable and lower-bounded for some $L \geq 0$.

**(C-2)** The stochastic gradient $G^t = \nabla f_{\xi_t}(W^{t-1})$ is an unbiased estimator of $\nabla f(W^{t-1})$ with bounded variance.

$$E_{\xi_t \sim \mathcal{D}}[G^t] = \nabla f(W^{t-1}), \quad E_{\xi_t \sim D}\left[\left\|G^t - \nabla f(W^{t-1})\right\|^2\right] \leq \frac{\sigma^2}{b_t}, \quad \forall t \geq 0,$$

where $b_t$ is the batch size of $\xi_t$ and $\sigma \geq 0$ is a constant.

**(C-3)** There are some $\rho_0, \mu \in [0, 1)$ and $D, G > 0$ such that $\|W^{t+1} - W^t\| \leq D$, $\|G^t\| \leq G$, and $\rho_t = \rho_0 \mu^{t-1}$ for all $t$.

**(C-4)** There is some $\gamma > 0$ such that $\|\eta_t^{-1}P^t\|_2 \leq 1/\gamma < \infty$ for all $t$.

**(C-5)** There is $\delta > 0$ such that

$$P^t \geq \delta, \quad \eta_t \leq \frac{\delta}{3L}, \quad \forall t \geq 0. \tag{31}$$

**Theorem 4.** *For the framework in [51] with the subproblem solved approximately by Algorithm 2 such that (7) holds with $\{\epsilon_t\}$ satisfying (8). Then Theorem 1 of [51] still holds, but with the constants $\{Q_i\}$ being also dependent on $\bar{\epsilon}$.*

*Proof.* The major flow of our proof follows that of [51] but with suitable modifications to accommodate the inexactness condition in the subproblem solving.

It is clear from [51, Lemma 1] that

$$\|m_t\| \leq G, \quad \forall t \geq 0. \tag{32}$$

By the update rule for $m_t$, (6) and (7), we have that there is $z_t$ such that

$$0 \in z_t + (1 - \rho_t)g_t + \rho_t m_{t-1} + \partial\psi(W^t) + \frac{1}{\eta_t}(P^t)(W^t - W^{t-1}), \quad \|z_t\| \leq \epsilon_t, \quad \forall t \geq 0,$$

leading to

$$\nabla f(W^t) - z_t - (1 - \rho_t)g_t - \rho_t m_{t-1} - \frac{1}{\eta_t}(P^t)(W^t - W^{t-1}) \in \partial F(W^t). \tag{33}$$

We thus have from (33) and (C-4) that

$\text{dist}(0, \partial F(W^t))^2$

$$\leq \left\| z_t + (1-\rho_t)g_t - \nabla f(W^t) + \rho_t m_{t-1} + (W^t - W^{t-1}) + \frac{1}{\eta_t}(P^t)(W^t - W^{t-1}) - (W^t - W^{t-1}) \right\|^2$$

$$\leq 4\left\| (1-\rho_t)g_t - \nabla f(W^t) + \rho_t m_{t-1} + (W^t - W^{t-1}) \right\|^2 + 4\epsilon_t^2 + 4\left\| \frac{1}{\eta_t}(P^t)(W^t - W^{t-1}) \right\|^2$$

$$+ 4\left\| (W^t - W^{t-1}) \right\|^2$$

$$\leq 4\underbrace{\left\| (1-\rho_t)g_t - \nabla f(W^t) + \rho_t m_{t-1} + (W^t - W^{t-1}) \right\|^2}_{T_1} + 4\left(\frac{1}{\gamma^2} + 1\right)\|W^t - W^{t-1}\|^2 + 4\epsilon_t^2.$$

$$(34)$$

We will separately bound the quantities $T_1$ and $\left\| W^t - W^{t-1} \right\|^2$ below.

From the subproblem objective requirement in (7), we get

$$\left\langle (1-\rho_t)g_t + \rho_t m_{t-1}, W^t - W^{t-1} \right\rangle + \psi(W^t) + \frac{1}{2\eta_t}\left\langle W^t - W^{t-1}, P^t(W^t - W^{t-1}) \right\rangle \leq \psi(W^{t-1}).$$

$$(35)$$

From (C-1), we have

$$f(W^t) \leq f(W^{t-1}) + \left\langle \nabla f(W^{t-1}), W^t - W^{t-1} \right\rangle + \frac{L}{2}\|W^t - W^{t-1}\|^2. \qquad (36)$$

Summing (35) and (36) gives

$$\left\langle (1-\rho_t)g_t - \nabla f(W^{t-1}) + \rho_t m_{t-1}, W^t - W^{t-1} \right\rangle + \left\| W^t - W^{t-1} \right\|^2_{\frac{P^t}{2\eta_t} - \frac{L}{2}I} \leq F(W^{t-1}) - F(W^t).$$

$$(37)$$

Note that $\eta_t^{-1}P^t - LI \succeq 0$ from (31), so the second term in (37) is nonnegative. (37) together with (C-3) then leads to

$\|W^t - W^{t-1}\|^2_{\frac{P^t}{2\eta_t} - \frac{L}{2}I}$

$$\leq F(W^{t-1}) - F(W^t) - \left\langle g_t - \nabla f(W^{t-1}), W^t - W^{t-1} \right\rangle + \left\langle \rho_t g_t, W^t - W^{t-1} \right\rangle - \left\langle \rho_t m_{t-1}, W^t - W^{t-1} \right\rangle$$

$$\leq F(W^{t-1}) - F(W^t) + \frac{1}{2L}\|g_t - \nabla f(W^{t-1})\|^2 + \frac{L}{2}\|W^t - W^{t-1}\|^2 + \frac{\rho_t^2}{2L}\|g_t\|^2 + \frac{L}{2}\|W^t - W^{t-1}\|^2$$

$$+ \rho_t\|m_{t-1}\|\|W^t - W^{t-1}\|$$

$$\leq F(W^{t-1}) - F(W^t) + \frac{1}{2L}\|g_t - \nabla f(W^{t-1})\|^2 + +L\|W^t - W^{t-1}\|^2 + \frac{\rho_0^2\mu^{2(t-1)}G^2}{2L} + \rho_0\mu^{t-1}DG.$$

Summing it over $t = 1, 2, \ldots, T$ and utilizing the assumption that the step sizes are non-increasing then give

$$\left(\frac{\delta}{2\eta_0} - \frac{3}{2}L\right)\sum_{t=1}^T \|W^t - W^{t-1}\|^2 \leq \Delta + C_1 + \frac{1}{2L}\sum_{t=1}^T \|g_t - \nabla f(W^{t-1})\|^2,$$

where

$$\Delta := F(W^0) - \min_W F(W), \quad C_1 := \frac{\rho_0 DG}{1-\mu} + \frac{\rho_0^2 G^2}{2L(1-\mu^2)}.$$

From the inequality above, we obtain

$$\sum_{t=1}^T \|W^t - W^{t-1}\|^2 \leq H_1 + H_2\sum_{t=1}^T \|g_t - \nabla f(W^{t-1})\|^2 \qquad (38)$$

for some constants $H_1, H_2$ depending on $L, \Delta, \delta, \eta_0$, and $C_1$. From (37), we also have

$\left\langle (1-\rho_t)g_t - \nabla f(W^t) + \rho_t m_{t-1}, W^t - W^{t-1} \right\rangle$

$$\leq F(W^{t-1}) - F(W^t) - \left\langle \nabla f(W^t) - \nabla f(W^{t-1}), W^t - W^{t-1} \right\rangle - \left\| W^t - W^{t-1} \right\|^2_{\frac{1}{2\eta_t}(P^t) - \frac{L}{2}I}$$

$$\leq F(W^{t-1}) - F(W^t) - \left\langle \nabla f(W^t) - \nabla f(W^{t-1}), W^t - W^{t-1} \right\rangle.$$

Therefore, we obtain

$$T_1 = \|(1-\rho_t)g_t - \nabla f(W^t) + \rho_t m_{t-1}\|^2 + \|W^t - W^{t-1}\|^2 + 2\Big\langle (1-\rho_t)g_t - \nabla f(W^t) + \rho_t m_{t-1}, W^t - W^{t-1}\Big\rangle$$

$$\leq \|(1-\rho_t)g_t - \nabla f(W^{t-1}) + \nabla f(W^{t-1}) - \nabla f(W^t) + \rho_t m_{t-1}\|^2 + \|W^t - W^{t-1}\|^2$$
$$\quad + 2\left(F(W^{t-1}) - F(W^t) - \langle \nabla f(W^t) - \nabla f(W^{t-1}), W^t - W^{t-1}\rangle\right)$$

$$\leq 4\|g_t - \nabla f(W^{t-1})\|^2 + 4L^2\|W^t - W^{t-1}\|^2 + 4\rho_t^2(\|m_{t-1}\|^2 + \|g_t\|^2) + \|W^t - W^{t-1}\|^2$$
$$\quad + 2\left(F(W^{t-1}) - F(W^t) + L\|W^t - W^{t-1}\|^2\right)$$

$$\leq 2\left(F(W^{t-1}) - F(W^t)\right) + 8\rho_0^2\mu^{2(t-1)}G^2 + \left(1 + 2L + 4L^2\right)\|W^t - W^{t-1}\|^2 + 4\|g_t - \nabla f(W^{t-1})\|^2. \tag{39}$$

Let $C_2 := 2 + 2L + 4L^2 + \gamma^{-2}$ and insert (39) into (34), we get

$$\text{dist}(0, \partial F(W^t))^2$$

$$\leq \ 4\left(2\left(F(W^{t-1}) - F(W^t)\right) + 8\rho_0^2\mu^{2(t-1)}G^2 + C_2\|W^t - W^{t-1}\|^2 + 4\|g_t - \nabla f(W^{t-1})\|^2 + \epsilon_t^2\right). \tag{40}$$

Therefore, we have from (8) and (40) and (C-2) that

$$\mathbb{E}_{a,\xi_1,\dots,\xi_T}[\text{dist}(0, \partial F(W^a))^2]$$

$$\leq \frac{1}{T}\sum_{t=1}^T \mathbb{E}\left[\Big\|(1-\rho_t)g_t - \nabla f(W^t) + z_t + \rho_t m_{t-1} + \frac{1}{\eta_t}(P^t)(W^t - W^{t-1})\Big\|^2\right]$$

$$\leq \frac{4}{T}\left(2\Delta + \frac{8\rho_0^2 G^2}{1-\mu^2} + 4\sum_{t=1}^T \mathbb{E}\|g_t - \nabla f(W^{t-1})\|^2 + C_2\sum_{t=1}^T \mathbb{E}\|W^t - W^{t-1}\|^2 + \sum_{t=1}^T \epsilon_t^2\right)$$

$$\leq \frac{4}{T}\left(2\Delta + \frac{8\rho_0^2 G^2}{1-\mu^2} + 4\sigma^2\sum_{t=1}^T \frac{1}{b_t} + C_2(H_1 + H_2\sigma^2\sum_{t=1}^T \frac{1}{b_t}) + \bar{\epsilon}\right)$$

$$\leq \frac{Q_1}{T}\sum_{t=1}^T \frac{1}{b_t} + \frac{Q_2\Delta}{T} + \frac{Q_3}{T},$$

for some constants $Q_1, Q_2, Q_3$ dependent on $\{\eta_0, \delta, \Delta, L, D, G, \rho_0, \mu, \gamma, \bar{\epsilon}\}$, but not on $T$. This proves our theorem. □

## C  Additional Experiments for Computer Vision

In this section, we compare RAMDA with other methods on image classification with smaller datasets. They are:

1. Logistic regression (neural network with no hidden layer) with the MNIST dataset [23].
2. A modified VGG19 [42] with the CIFAR10 dataset [21].
3. The same VGG19 with the CIFAR100 dataset [21].
4. A modified ResNet50 [14] with the CIFAR10 dataset.
5. The same ResNet50 with the CIFAR100 dataset.

The results are shown in Table 7. In the logistic regression problem, we only perform a single run, with the initial point being the origin, as it is a convex problem. Moreover, in this problem, when dealing with ProxSSI, ProxGen, and ProxSGD whose sparsity levels are highly unstable over iterations, we report their highest weighted group sparsity over all epochs, but for all other problems, we report the group sparsity level of the final output.

Experiments in this subsection show that RAMDA might sometimes perform worse than existing methods on smaller problems like CIFAR10/100. Fortunately, for such smaller problems, the training cost is not very significant, and one can afford to try more algorithms.

Table 7: Group sparsity and validation accuracy of different methods on image classification with smaller datasets.

| Algorithm | Validation accuracy | Group sparsity |
|---|---|---|
| Logistic Regression/MNIST | | |
| ProxSGD | 91.31% | 39.29% |
| ProxSSI | 91.31% | 39.92% |
| ProxGen | 91.31% | 39.92% |
| RMDA | 91.34% | 57.02% |
| RAMDA | **91.35%** | **57.40%** |
| VGG19/CIFAR10 | | |
| MSGD | $93.95 \pm 0.14\%$ | - |
| ProxSGD | $92.82 \pm 0.09\%$ | $82.76 \pm 5.42\%$ |
| ProxSSI | $92.81 \pm 0.15\%$ | $88.40 \pm 0.23\%$ |
| ProxGen | $92.83 \pm 0.05\%$ | $86.64 \pm 0.12\%$ |
| RMDA | $\mathbf{93.13 \pm 0.10\%}$ | $\mathbf{90.22 \pm 0.06\%}$ |
| RAMDA | $92.89 \pm 0.13\%$ | $86.31 \pm 0.31\%$ |
| VGG19/CIFAR100 | | |
| MSGD | $74.07 \pm 0.05\%$ | - |
| ProxSGD | $\mathbf{71.96 \pm 0.15\%}$ | $72.34 \pm 11.9\%$ |
| ProxSSI | $67.29 \pm 0.06\%$ | $78.58 \pm 0.34\%$ |
| ProxGen | $68.13 \pm 0.36\%$ | $75.46 \pm 0.17\%$ |
| RMDA | $\mathbf{71.96 \pm 0.31\%}$ | $\mathbf{80.88 \pm 0.11\%}$ |
| RAMDA | $70.47 \pm 0.25\%$ | $65.19 \pm 0.77\%$ |
| ResNet50/CIFAR10 | | |
| MSGD | $95.54 \pm 0.19\%$ | - |
| ProxSGD | $92.36 \pm 0.05\%$ | $82.18 \pm 2.67\%$ |
| ProxSSI | $94.04 \pm 0.12\%$ | $83.67 \pm 0.63\%$ |
| ProxGen | $94.07 \pm 0.12\%$ | $80.45 \pm 0.45\%$ |
| RMDA | $\mathbf{95.11 \pm 0.11\%}$ | $\mathbf{85.64 \pm 0.12\%}$ |
| RAMDA | $93.85 \pm 0.10\%$ | $81.99 \pm 1.26\%$ |
| ResNet50/CIFAR100 | | |
| MSGD | $79.49 \pm 0.49\%$ | - |
| ProxSGD | $74.54 \pm 0.58\%$ | $49.29 \pm 5.91\%$ |
| ProxSSI | $73.65 \pm 0.39\%$ | $70.38 \pm 0.74\%$ |
| ProxGen | $73.63 \pm 0.43\%$ | $65.51 \pm 3.58\%$ |
| RMDA | $\mathbf{75.62 \pm 0.19\%}$ | $\mathbf{79.97 \pm 0.27\%}$ |
| RAMDA | $69.23 \pm 0.86\%$ | $68.65 \pm 1.83\%$ |

## D  Plots of Sparsity Level and Validation Accuracy over Epochs

We provide in Fig. 1 the plots of predictive ability and structured sparsity over epochs of some representative experiments we have conducted. These experiments are:

1. ResNet50 with the ILSVRC 2012 ImageNet dataset.
2. Transformer-XL with the WikiText-103 dataset.
3. Tacotron2 with the LJSpeech dataset.
4. Logistic Regression with the MNIST dataset.
5. A modified VGG19 with the CIFAR10 dataset.
6. The same VGG19 with the CIFAR100 dataset.
7. A modified ResNet50 with the CIFAR10 dataset.
8. The same ResNet50 with the CIFAR100 dataset.

In the plot for Transformer-XL, one step processes ten batches, and for our batch size of 64, one epoch consists of 8,401 batches. We further observe in the zoomed-in sparsity

plots in Fig. 2 that the sparsity level of RAMDA is stable at the final epochs. These plots corroborates our theory that RAMDA is indeed capable of manifold identification while achieving competitive prediction performance. On the other hand, in the absence of manifold identification guarantees, the sparsity levels of ProxSGD, ProxSSI and ProxGen exhibit oscillations that are sometimes drastic. We note that for the largest problems Tacotron2 and Transformer-XL, the sparsity levels of RAMDA are still gradually increasing even at the final epochs. This suggests that if we are willing to run the algorithm for longer, it is possible that the structured sparsity level could be further improved.

## E Experiment with Nuclear-norm Regularization

We further conduct some preliminary experiments with a different regularizer to showcase that the proposed RAMDA can be applied to structures beyond sparsity. We consider the structure such that each layer of the neural network is low-rank, induced by imposing one nuclear-norm regularizer per layer individually by treating each layer as a matrix. Given a matrix $X \in \mathbb{R}^{m \times n}$ of rank $r \leq \min\{m, n\}$ with its singular value decomposition (SVD) $X = U\Sigma V^\top$, where $U \in \mathbb{R}^{m \times r}$, $V \in \mathbb{R}^{n \times r}$ are orthogonal and the positive definite diagonal matrix $\Sigma \in \mathbb{R}^{r \times r}$ represents the nonzero singular values of $X$, the nuclear norm of $X$ is computed by

$$\|X\|_* = \sum_{i=1}^{r} \Sigma_{i,i},$$

and the corresponding proximal operator for $\lambda > 0$ is

$$\text{prox}_{\lambda \|\cdot\|_*}(X) = U\hat{\Sigma}V^\top, \text{ where } \hat{\Sigma}_{i,i} = \max\{0, \Sigma_{i,i} - \lambda\}, \quad i = 1, \ldots, r.$$

Given a point $X^*$ with rank $r^*$, the active manifold of the nuclear norm at $X^*$ is

$$\mathcal{M}(X^*) = \{Y \mid \text{rank}(Y) = r^*\}.$$

Using low-rankness to condense neural networks is itself an interesting research topic, but conducting full SVDs could be rather time-consuming, so applying this structure to larger problems is challenging but potentially useful. How to exploit this structure for prediction acceleration and to make the training more efficient, possibly using iterative methods to compute approximate SVDs, is an interesting topic we plan to investigate in the near future. Instead, the purpose of the preliminary experiment here is merely for showing that our method is also applicable to other structures.

We first consider training a simple neural network with six fully-connected layers using the FashionMNIST dataset [46]. Since this is a rather small-scale problem and this is a image classification problem, we do not expect RAMDA to outperform non-adaptive methods, especially the RMDA method that is also able to identify the active manifold. The goal of this experiment is just to demonstrate the possibilities of structures beyond sparsity. The results are shown in Table 8. As we have anticipated, RAMDA is indeed slightly worse than RMDA regarding the low-rank level and the prediction accuracy, but it is still competitive and outperforms ProxGen and ProxSGD. This exemplifies the potential of RAMDA as well as RMDA for training neural networks with other useful structures.

We also conduct an experiment on pretraining a modified vision transformer model [29] for masked image modeling [48] using the CIFAR10 dataset. Following the standard practice of this task, we select the model that gives the lowest validation loss among the last 50 epochs as the final output. The results are shown in Table 9. We can see that RAMDA attains the lowest validation loss and has a low-rank level almost identical to that of RMDA. On the other hand, ProxSGD and ProxGen have worse low-rank levels.


Figure 1: Group sparsity level and validation prediction performance v.s epochs. In the plot for Transformer-XL, one step processes ten batches, and for our batch size of 64, one epoch consists of 8,401 batches.

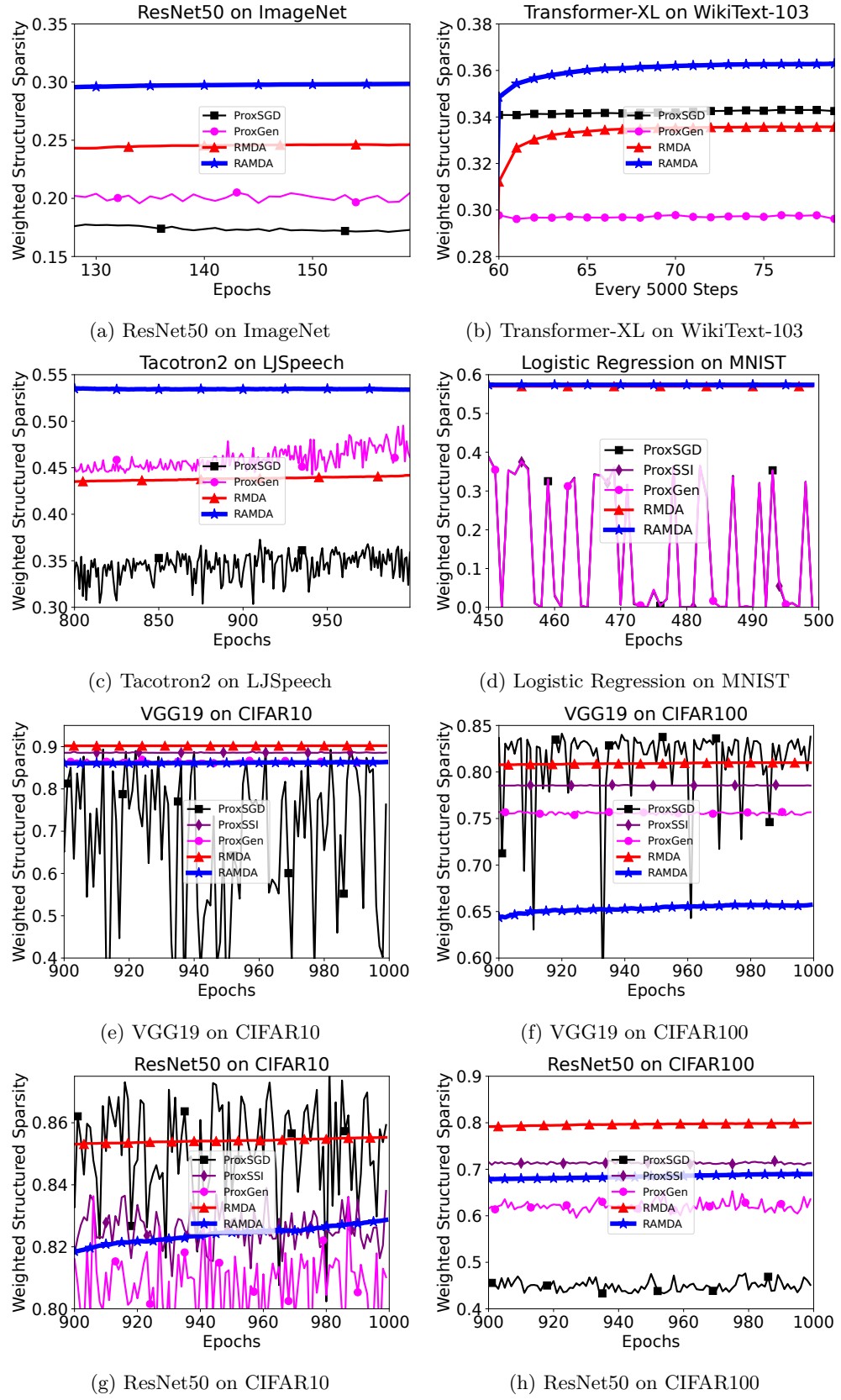

Figure 2: Group sparsity level at the last epochs.

Table 8: Low-rank level and validation accuracy of different methods on training a six-layer fully-connected neural network with the FashionMNIST dataset for image classification.

| Algorithm | Validation accuracy | Low-rank level |
|---|---|---|
| MSGD | $89.95 \pm 0.29\%$ | - |
| ProxSGD | $87.54 \pm 0.52\%$ | $78.00 \pm 0.77\%$ |
| ProxGen | $86.66 \pm 0.33\%$ | $87.46 \pm 4.19\%$ |
| RMDA | $\mathbf{88.19 \pm 0.23\%}$ | $\mathbf{91.88 \pm 0.12\%}$ |
| RAMDA | $87.99 \pm 0.24\%$ | $89.59 \pm 0.42\%$ |

Table 9: Low-rank level and validation loss of different methods on pretraining a modified vision transformer model using the CIFAR10 dataset for masked image modeling.

| Algorithm | Validation loss | Low-rank level |
|---|---|---|
| AdamW | $0.0865 \pm 0.0001$ | - |
| ProxSGD | $0.1042 \pm 0.0003$ | $82.60 \pm 0.34\%$ |
| ProxGen | $0.1120 \pm 0.0019$ | $82.64 \pm 2.47\%$ |
| RMDA | $0.1054 \pm 0.0031$ | $\mathbf{86.23 \pm 0.41\%}$ |
| RAMDA | $\mathbf{0.1035 \pm 0.0016}$ | $\mathbf{86.20 \pm 0.35\%}$ |

Answer: [Yes]

Justification: Our claims accurately reflect the paper's contributions and scope.

2. **Limitations**

Question: Does the paper discuss the limitations of the work performed by the authors?

Answer: [Yes]

Justification: Limitations of our work are discussed in Section 5 (after Theorem 2) and Appendix C.

3. **Theory Assumptions and Proofs**

Question: For each theoretical result, does the paper provide the full set of assumptions and a complete (and correct) proof?

Answer: [Yes]

Justification: All assumptions are stated clearly in each theorem statement, and all detailed proofs are provided in Appendix B.

4. **Experimental Result Reproducibility**

Question: Does the paper fully disclose all the information needed to reproduce the main experimental results of the paper to the extent that it affects the main claims and/or conclusions of the paper (regardless of whether the code and data are provided or not)?

Answer: [Yes]

Justification: Algorithm details are all given in the paper, and the parameter settings are all available in the supplementary materials.

5. **Open access to data and code**

Question: Does the paper provide open access to the data and code, with sufficient instructions to faithfully reproduce the main experimental results, as described in supplemental material?

Answer: [Yes]

Justification: Our code is provided in the supplementary materials, our data are public data sets, and sufficient instructions are given in the README in the supplementary materials.

6. **Experimental Setting/Details**

   Question: Does the paper specify all the training and test details (e.g., data splits, hyperparameters, how they were chosen, type of optimizer, etc.) necessary to understand the results?

   Answer: [Yes]

   Justification: All details are given in either the main paper or the supplementary materials.

7. **Experiment Statistical Significance**

   Question: Does the paper report error bars suitably and correctly defined or other appropriate information about the statistical significance of the experiments?

   Answer: [Yes]

   Justification: We report the mean and standard deviation of the comparison criteria over three different random initializations.

8. **Experiments Compute Resources**

   Question: For each experiment, does the paper provide sufficient information on the computer resources (type of compute workers, memory, time of execution) needed to reproduce the experiments?

   Answer: [Yes]

   Justification: Time of execution is reported in Section 6, and details of the computer resources are given in Appendix A.3.

9. **Code Of Ethics**

   Question: Does the research conducted in the paper conform, in every respect, with the NeurIPS Code of Ethics https://neurips.cc/public/EthicsGuidelines?

   Answer: [Yes]

   Justification: The research conducted in the paper conform, in every respect, with the NeurIPS Code of Ethics.

10. **Broader Impacts**

    Question: Does the paper discuss both potential positive societal impacts and negative societal impacts of the work performed?

    Answer: [NA]

    Justification: This is a fundamental research work and there is no foreseeable negative societal impact.

11. **Safeguards**

    Question: Does the paper describe safeguards that have been put in place for responsible release of data or models that have a high risk for misuse (e.g., pretrained language models, image generators, or scraped datasets)?

    Answer: [NA]

    Justification: The paper poses no such risks.

12. **Licenses for existing assets**

    Question: Are the creators or original owners of assets (e.g., code, data, models), used in the paper, properly credited and are the license and terms of use explicitly mentioned and properly respected?

    Answer: [Yes]

    Justification: We do cite all of the papers that proposed the models, datasets, and code we used.

13. **New Assets**

    Question: Are new assets introduced in the paper well documented and is the documentation provided alongside the assets?

Answer: [Yes]

Justification: This paper will introduce new open-source software, and we have provided a README file for documentation of the package.

14. **Crowdsourcing and Research with Human Subjects**

Question: For crowdsourcing experiments and research with human subjects, does the paper include the full text of instructions given to participants and screenshots, if applicable, as well as details about compensation (if any)?

Answer: [NA]

Justification: The paper does not involve crowdsourcing nor research with human subjects.

15. **Institutional Review Board (IRB) Approvals or Equivalent for Research with Human Subjects**

Question: Does the paper describe potential risks incurred by study participants, whether such risks were disclosed to the subjects, and whether Institutional Review Board (IRB) approvals (or an equivalent approval/review based on the requirements of your country or institution) were obtained?

Answer: [NA]

Justification: The paper does not involve crowdsourcing nor research with human subjects.

