# OpenReview forum: "Regularized Adaptive Momentum Dual Averaging with an Efficient Inexact Subproblem Solver for Training Structured Neural Network"
_NeurIPS.cc/2024/Conference — NeurIPS 2024 poster_

### Official Review · Reviewer_4z2u · 2024-07-11

**Soundness:** 3
**Presentation:** 3
**Contribution:** 3
**Rating:** 5
**Confidence:** 4

**Summary:**

This article proposes an optimization algorithm RAMDA for training structured neural networks, which combines a number of optimization techniques including dual averaging, momentum, and coordinate-wise preconditioners. Similar to the existing RMDA algorithm, RAMDA also has the capacity to identify the local manifold structure of the solution. The author(s) provide theoretical analyses to justify the convergence property of RAMDA, and develop an inexact subproblem solver as required by RAMDA.

**Strengths:**

The proposed RAMDA algorithm extends the existing RMDA algorithm by adding a coordinate-wise preconditioner, and its theoretical analysis seems to be novel.

**Weaknesses:**

I think one major weakness of the current manuscript is the **correctness** of some theoretical results presented in the article.

1. Theorem 1 suggests that the regularizer function $\psi$ can be nonconvex. However, as the RAMDA algorithm heavily relies on the proximal operator of $\psi$, how do you define the proximal operator when $\psi$ is nonconvex? For example, equation (6) is used to define the new iterate $W^t$, but when $\psi$ is nonconvex, it is likely that the "argmin" is a set and is not uniquely defined.

2. Taking a closer look at the proof of Theorem 1, I feel that the author(s) may have a misunderstanding of an existing theorem. In Appendix B, equation (11) is obtained by citing Theorem 10.15 of [1]. However, Theorem 10.15 of [1] applies to functions of the form $F(x)=f(x)+\psi(x)$, where $f$ is smooth and nonconvex, but $\psi$ is convex. In other words, the non-convexity only applies to the smooth part, not the regularizer.

3. If the findings above are valid, then the author(s) may need a thorough examination of the technical proofs to see if there is any error.

4. If we assume $\psi$ is convex, then there should be an Nesterov-accelerated version of Algorithm 2 that converges in $O(\varepsilon_t^{-1/2})$ iterations, which is faster than the rate given in Theorem 1.

[1] Beck, A. (2017). First-order methods in optimization. Society for Industrial and Applied Mathematics.


===============================================================

Edit: during the rebuttal the author(s) seem to have addressed the concerns above.

**Questions:**

See the "Weaknesses" section.

**Limitations:**

The authors have discussed the limitations of the proposed algorithm in Section 5 and Appendix C.

---

> ### Author Rebuttal · Authors · 2024-08-06
>
> We would like to thank the reviewer for the careful checking of our paper, including the proofs.
> In current days, it has been increasingly rare to see a reviewer spending this much effort and time to review papers, and we truly appreciate it.
> And we thank the reviewer for pointing out our careless errors, which provides us a chance to correct them and obtain better guarantees.
> Our reply is as follows.
>
> **Q1**
>
> It is indeed correct that when the regularizer is nonconvex, the argmin could be a set, and thus the proximal operator becomes a set operator.
> This is already reflected in our proof of Thm 1 (see the eq between Lines 544 and 545 on P.14).
> In this case, we can select any minimizer from the set of minima and the analysis would not be affected. (In the main paper, since we are only approximately solving the problem, whether the real solution is unique does not matter in the algorithmic description.)
> This is the standard case in analysis involving the proximal operator of a nonconvex function. See, for example, the canonical work of Attouch et al. (2013).
>
> **Q2 & Q3**
>
> Our result is correct, but indeed the citation is incorrect and one minor assumption is missing, and thank you for pointing it out.
> A better reference (although without a formal theorem but just some discussion) would be section 5.1 of Attouch et al. (2013).
> And the additionally required assumption is that the objective function is lower-bounded by some finite value (which does not necessarily require convexity of the regularizer).
> For completeness, we provide a proof for the claimed rate in (11) here.
>
> Consider applying proximal gradient with stepsize $\theta_t$ to a function $\bar Q_t(Z) := f_t(Z) + \psi (Z)$, where $f_t$ is $L$-smooth and $\psi$ comes with an easily computable proximal operator.
> Let the iterates be updated by
>
> $$Z^{j+1} \in \arg\min_{Z} \left(\hat Q_t(Z; Z^j) := \langle \nabla f_t(Z^j), Z - Z^j \rangle + \frac{1}{2 \theta_t} \|Z - Z^j \|^2 + \psi(Z)\right),$$
>
> then from that $Z^{j+1}$ is a minimizer of $\hat Q_t(Z; Z^j)$, we know that
> $\hat Q_t(Z^{j+1}; Z^j) \leq \hat Q_t(Z^j; Z^j)$, which implies
>
> $$\langle \nabla f_t(Z^j), Z^{j+1} - Z^j \rangle + \frac{1}{2 \theta_t} \|Z^{j+1} - Z^j \|^2 + \psi(Z^{j+1}) \leq \psi(Z^j).$$
>
> On the other hand, from the $L$-smoothness of $f_t$, we know that
>
> $$f_t(Z^{j+1}) - f_t(Z^j) \leq \langle \nabla f_t(Z^j), Z - Z^j \rangle + \frac{L}{2} \|Z - Z^j \|^2.$$
>
> Adding these two inequalities together, we obtain that
>
> $$\bar Q_t(Z^j) - \bar Q_t(Z^{j+1}) \geq (\frac{1}{2 \theta_t} - \frac{L}{2}) \|Z^{j+1} - Z^j\|^2.$$
>
> By taking $\theta_t = 1/(2L)$ (as we did in the first line of Alg 2), summing the above inequality for $j=0,1,2,\dotsc,T$, and noting that $\bar Q_t(Z^0) - \bar Q_t(Z^{T+1})$ is upper bounded by $\bar Q_t(Z^0) - \bar Q_t^*$, we obtain our claimed rate in equation (11).
>
> **Q4**
>
> For the convex case, now we realize that the rate can actually be improved to $O(\log \epsilon_t^{-1})$. The idea is that when the regularizer is convex, $\hat Q_t$ is actually strongly convex.
> Indeed, from our construction of $P^t$, we know that the smallest diagonal entry is at least $\epsilon$, and therefore the subproblem is $\epsilon$-strongly convex and the smooth part is $L$-smooth, where $L = \max(diag(P^t)) + \epsilon$.
> We denote the global optimum by $\bar Q^*$ and $\kappa := L/\epsilon$.
> We know that $\bar Q_t(Z^j) - \bar Q_t(Z^{j+1})$ is upper bounded by $\bar Q_t(Z^j) - \bar Q_t^*$, which converges at a linear rate (see, for example, equation (3.4) of Drusvyatskiy & Lewis (2018)):
>
> $$ \bar Q_t(Z^j) - \bar Q_t (Z^{j+1}) \leq \bar Q_t(Z^j) - \bar Q_t^* \leq \left(1 - \frac{1}{2\kappa}\right)^j(\bar Q_t(Z^0) - \bar Q_t^*).$$
>
> On the other hand, with the stepsize being $1/L$, we know from Theorem 1 of Nesterov (2013) that
> $$\bar Q_t(Z^j) - \bar Q_t(Z^{j+1}) \geq \frac{L}{2} \|Z^j - Z^{j+1}\|^2.$$
> Combination of these two results then gives that it takes at most $O(\log \epsilon_t^{-1})$ iterations to satisfy (7).
>
> On the other hand, for a convex $\psi$, accelerated proximal gradient has the same dependency on $\epsilon_t$, although with a possibly faster linear convergence rate.
> However, accelerated proximal gradient does not guarantee that the objective value is always decreasing, so the second condition in (7) might be violated.
> This is the why we did not apply it as our subproblem solver.
>
>
> **References:**
>
> Attouch, Hedy, Jérôme Bolte, and Benar Fux Svaiter. "Convergence of descent methods for semi-algebraic and tame problems: proximal algorithms, forward–backward splitting, and regularized Gauss–Seidel methods." Mathematical Programming 137.1 (2013): 91-129.
>
> Nesterov, Yu. "Gradient methods for minimizing composite functions." Mathematical programming 140.1 (2013): 125-161.
>
> Drusvyatskiy, Dmitriy, and Adrian S. Lewis. "Error bounds, quadratic growth, and linear convergence of proximal methods." Mathematics of Operations Research 43.3 (2018): 919-948.

---

> > ### Comment · Reviewer_4z2u · 2024-08-08
> >
> > Thanks for the updates. I'll need some time reading the new materials.

---

> > > ### Author Response · Authors · 2024-08-11
> > > **Typos**
> > >
> > > We would like to correct two typos in our previous rebuttal.
> > > 1. In our reply to Q2 & 3, in the one line inequality between "On the other hand ......" and "Adding these two ......", the two $Z$ on the right-hand side should be $Z^{j+1}$.
> > >
> > > 2. In our reply to Q4, $L$ should be defined as $L =max(diag(P^t))$. ($\epsilon$ is already included in the definition of $P^t$).
> > >
> > > Sorry for the confusions these typos might have been caused.

---

> > > > ### Comment · Reviewer_4z2u · 2024-08-13
> > > >
> > > > I thank the author(s) for the detailed response. My previous concerns seem to have been addressed, and hence I would like to raise my score.

---

### Official Review · Reviewer_xuwn · 2024-07-12

**Soundness:** 3
**Presentation:** 3
**Contribution:** 3
**Rating:** 7
**Confidence:** 3

**Summary:**

This paper develops regularized adaptive momentum dual averaging (RAMDA) for structured neural networks. The method uses the preconditioning matrix to accelerate the convergence of a regularized momentum dual averaging (RMDA) method at the price of requiring the local solver (e.g. standard proximal gradient methods) to solve the subproblem. By the preconditioning matrix inspired by the AdaGrad stepsizes in Eq. (2), RAMDA outperforms RMDA and other existing gradient-based methods for solving structured neural networks in various learning applications.

**Strengths:**

1. Theoretical results suggest the convergence towards the solution to the subproblem when the proximal gradient methods are used as the local solvers, and almost surely convergence of RAMDA that derives from the manifold theory under the standard $L$-smoothness assumption on the objective functions $f$.

2. Empirical results illustrate the superior performance of RAMDA over RMDA and other existing gradient-based methods for various neural network tasks. Clear criteria, e.g. for solving the subproblems, are clearly stated in the numerical experiments.

**Weaknesses:**

1. I think there is an error in Eq. (3) where it should be the square root $\sqrt{\cdot}$ in the diagonal operator for $P^t$. This is because $P^t$ uses $U^t$ that is computed from the element-wise multiplicative product of the gradient $G^t$. Is $P^t$ inspired by the AdaGrad stepsizes? If so, then adding the justifications on using $P^t$ in RAMDA is worthwhile to better distinct RAMDA from RMDA.
2. In the experiments, can you comment on the impact of the different $\epsilon_t$ on the training performance of RAMDA? Because I believe that using $\epsilon_t$ a bit higher than $10^{-8}$ set in your experiments RAMDA might achieve far lower training time than other methods while keeping still comparable perplexity to solve Transformer-XL with WikiText-103 in Table 4, or Tacotron2 with LJSpeech in Table 5.

**Questions:**

I listed questions as part of weaknesses.

**Limitations:**

The limitations of current theoretical results are clearly and fairly discussed after Theorem 2.

---

> ### Author Rebuttal · Authors · 2024-08-06
>
> We would like to thank the reviewer for the careful evaluation of our paper and the invaluable suggestions. Our response is as follows.
>
> **Q1.**
>
>  Equation 3 is indeed correct that we are using the cube root.
>   This choice of the preconditioner follows the empirical success of the MADGRAD algorithm of Defazio & Jelassi (2022) for smooth
>   optimization, and our proof also indicates that using the cube root still leads to convergence.
>   More discussion about the reasoning of using the cube root instead can be found in Sec 5.5.4 of Defazio & Jelassi (2022), but in
>   general we consider that the numerical results as more strong evidence than those verbal explanations.
>   On the other hand, even using the cube root, $P^t$ is indeed still inspired by AdaGrad to accumulate the previous gradient norm
>   squared. We will add discussion about AdaGrad in our revision
>
> **Q2.**
>
> It is expected that with a looser subproblem stopping condition, we could reduce some running time of RAMDA. However, it is quite
>   unlikely that it can be faster than other methods: all methods share some common operations including calculation of the gradient
>   (which is usually the major bottleneck) and several vector additions (for updating the iterates and the accumulated gradient norm) per
>   iteration, and using a looser subproblem stopping condition would not affect these parts. The best we can hope for is that RAMDA and
>   ProxGen would be no slower than other methods that do not involve solving a subproblem with no closed-form solution.
>
>   With further investigation, we found that In Tacotron2 with LJSpeech, the time spent on solving the subproblem is very little, at 2.7%
>   and 1.6% of the total training time for RAMDA and ProxGen, respectively.
>   This suggests that loosening the subproblem stopping condition would provide only very limited benefit.
>   For Transformer-XL with WikiText-103, the time percentages spent on solving the subproblems are 10% for RAMDA and 5.9% for
>   ProxGen, indicating a rather high burden.
>   Consequently, for this problem, we tried to increase the stopping threshold from $10^{-8}$ to $10^{-6}$, which then resulted in a 20.2%
>   reduction in the subproblem time for RAMDA, while that of ProxGen almost remained unchanged (which suggests that probably there
>   is a sharp decrease in the objective improvement so that it reached both the old and the new stopping conditions at the same
>   iteration). As you conjectured, Both RAMDA and ProxGen with this looser stopping condition produced validation perplexity and
>   weighted structured sparsity comparable the original ones. However, as mentioned above, the running time difference is actually not that huge. (For ProxGen it is  likely that the algorithm behavior actually did not change given that the subproblem time remains unchanged.)

---

> > ### Comment · Reviewer_xuwn · 2024-08-12
> >
> > Thank you for your detailed response, and your empirical results with fair discussions on the impact of $\epsilon_t$. Because of this, I would like to maintain my score.

---

### Official Review · Reviewer_X8bY · 2024-07-13

**Soundness:** 3
**Presentation:** 3
**Contribution:** 3
**Rating:** 7
**Confidence:** 3

**Summary:**

#### Summary
The paper introduces the Regularized Adaptive Momentum Dual Averaging (RAMDA) algorithm for training structured neural networks. RAMDA addresses the challenge of solving the subproblem involved in the regularized adaptive methods, which typically lacks a closed-form solution. The paper presents an inexactness condition that retains convergence guarantees and proposes an efficient subproblem solver. The algorithm leverages manifold identification theory to ensure that the iterates of RAMDA attain the ideal structure induced by the regularizer at convergence. Extensive experiments demonstrate the effectiveness of RAMDA in various tasks, including computer vision, language modeling, and speech synthesis.

**Strengths:**

#### Strengths
1. **Novel Algorithm**: RAMDA combines adaptive momentum dual averaging with efficient inexact subproblem solving, providing a practical and theoretically sound method for training structured neural networks.
2. **Theoretical Guarantees**: The paper provides strong theoretical support, including convergence guarantees and structure identification, ensuring the algorithm's robustness.
3. **Practical Efficiency**: The proposed inexact subproblem solver is efficient, making RAMDA feasible for large-scale applications.
4. **Empirical Validation**: Extensive experiments across multiple domains demonstrate the superior performance of RAMDA in terms of both prediction accuracy and structured sparsity.

**Weaknesses:**

#### Weaknesses
1. **Computational Complexity**: The computational complexity of the proposed subproblem solver, especially for high-dimensional data, needs more detailed discussion.
2. **Generality**: While the paper focuses on specific types of structured neural networks, extending the methodology to other models and regularizers would enhance its generality.
3. **Comparative Analysis**: More detailed comparisons with other state-of-the-art methods, beyond the provided benchmarks, would strengthen the empirical validation.
4. **Implementation Details**: Practical guidelines for implementing RAMDA, including parameter tuning and handling different data distributions, are somewhat lacking.

**Questions:**

#### Questions
1. **Computational Complexity**:
    - Could you provide more details on the computational complexity of the proposed subproblem solver? How does it scale with increasing data size and model complexity?

2. **Generality**:
    - The paper focuses on structured neural networks with specific regularizers. Are there any challenges in extending RAMDA to other types of models or regularizers, such as those used in different machine learning tasks?

3. **Comparison with Existing Methods**:
    - How does RAMDA compare empirically with other state-of-the-art methods for training structured neural networks? Are there specific scenarios where RAMDA significantly outperforms these methods?

4. **Implementation Guidelines**:
    - Can you offer practical guidelines for implementing RAMDA in real-world scenarios? Specifically, how should practitioners tune the parameters, such as the learning rate and the inexactness threshold?

5. **Assumptions and Limitations**:
    - The paper discusses some assumptions and limitations. Could you elaborate on the key assumptions that are critical for the theoretical results, and how robust the method is to violations of these assumptions?

---

> ### Author Rebuttal · Authors · 2024-08-07
>
> We thank the reviewer for the evaluation of our paper. Our reply is as follows.
>
> 1. Computational Complexity: The complexity of the subproblem (Eq. 4) depends on the regularizer and especially its associated
>     proximal operation. Let the subproblem dimension, which is also the model size, be $n$. Most of the widely used regularizers come
>     with a proximal operator whose cost scales at $O(n)$, like the sparsity-inducing ones. However, there are also cases with higher
>     dependency on the problem dimension, such as the nuclear norm regularizer we adopted in appendix E, whose cost scales at
>     $O(n^3)$. Therefore, the subproblem complexity dependency on the model size would be dependent on the regularizer.
>     On the other hand, the data size does not affect the subproblem solver at all -- likely it affects the number of outer iterations and the
>     time cost of computing the stochastic gradient.
>
> 2. Generality: We disagree that this paper focused on specific regularizers. Indeed for experimental purposes we have to select some
>     representative regularizers because there is no way to exhaust all possible regularizers in experiments. However, our discussion,
>     assumption on the regularizer, and analysis are all general such that a broad class of regularizers are included. On the other hand,
>     as we discussed in Appendix E and in the previous point, some regularizers might be useful but too costly for the subproblem solver,
>     and it might require more domain knowledge to design better structure grouping or subproblem solvers to tackle this cost issue in
>     these cases.
>
>
>     As for extending to other machine learning tasks, RAMDA is specifically designed for deep neural networks and probably less
>     suitable for other machine learning tasks. In particular, adaptive models are not that widely used in other machine learning tasks, up
>     to our knowledge, and therefore we do not expect our method to be as competitive on other machine learning tasks.
>
>
> 3. Comparison with Existing Methods: We have compared RAMDA with other state-of-the-art methods for training structured neural networks in Section 6. Our numerical results have clearly indicated that RAMDA produces better prediction performance and higher structuredness simultaneously than state of the art, and with running time comparable to the most efficient algorithm. This shows that RAMDA indeed pushes forward the state of the art for training structured neural networks.
> It is well-known that adaptive methods tend to be most successful in many state-of-the-art deep neural networks, especially the transformer (see Zhang et al. (2020)). It is also the case for RAMDA: our experiments have shown that it performs the best on such models. It also has outstanding performance on large-scale problems like the ImageNet problem.
>
> 4. Implementation Guidelines:
>     There are four type of possible hyperparameters to tune:
>
>     - Learning rate: We do grid search to tune this hyperparameter in all of our experiments. Otherwise, $10^{-2}$ is a good guess if the
>         computational resources are limited.
>
>     - Learning rate scheduling: In all of the experiments, we use stage-wise learning rate scheduling for RAMDA. Each stage has an equal number of iterations, and the multiplicative factor of learning rate decay at each stage is $10^{-1}$.
>
>     - $\lambda$ (weight of the regularization): We do grid search to tune this hyperparameter in all of our experiments.
>
>     - Momentum: $10^{-1}$ or $10^{-2}$ are good choices. We simply chose $10^{-2}$ in all the experiments of the Section 6.
>
>     Inexactness threshold: There are two hyperparemeters to controll the inexactness threshold (see Section 6):
>
>     - max_iters: the maximun number of the iterations. We set it to $100$ in all the experiments.
>
>     - rtol (: (previous_subproblem_objective-current_subproblem_objective)/(abs(current_subproblem_objective)+1.0): criterion for early
>         stopping. We set it to $10^{-8}$ in all the experiments except Table 1.
>
>     Implementation and experimental code for reproducibility has also been included in the supplementary materials.
>
> 5. Assumptions and Limitations: Thms 1 & 4 have very little assumptions and limitations, and therefore they are applicable to almost all situations. For Thm 2, as discussed after the theorem statement, the key assumption is that the iterates converge to a point. Its satisfaction might require some additional regularity conditions that are usually satisfied by generic problems, but the analysis would be very involved and we will leave it as future work. In practice, we notice that with proper hyperparameter search, this assumption seems to hold true in general -- usually those hyperparameters that lead to divergent iterates are not adopted.
> For Thm 3, the nondegeneracy condition is the key assumption, and again it holds for generic problems (since the relative boundary is a measure zero set, that the negative gradient falls on it is a probability zero event.)

---

> > ### Author Response · Authors · 2024-08-08
> > **Learning rate scheduling setup**
> >
> > More details regarding the learning rate scheduling setup: We begin with determining the total number of epochs, often considered as a budget, and then split them into 3 to 5 roughly equal stages. While more epochs typically lead to better performance when resources allow, in some cases, the total number of epochs should be adjusted based on validation performance to identify the optimal point.

---

### Author Rebuttal · Authors · 2024-08-06

We thank the reviewers for their careful evaluations of our paper.
We will individually reply to the reviewers to address their specific questions, and here we would like to highlight some changes that we will make in our revision.

- For Theorem 1, the rate when $\psi$ is convex can be improved to $O(\log \epsilon_t^{-1})$. This is from the unutilized observation that the objective function is actually strongly convex, and thus proximal gradient is expected to have a linear rate of convergence instead of the current slower sublinear one.

- In the proof of Theorem 1, reviewer 4z2u pointed out that there is a wrong citation for the case of nonconvex $\psi$. A better reference would be Section 5.1 of Attouch et al. (2013). We will also add the derivation to show that indeed this rate is correct, just the citation was wrong. More details are in our response to reviewer 4z2u.

- There is a typo in our proof of Theorem 1: in (16), (21), and the unnumbered
  equation before (21),

$$\nabla E_{\xi \sim D} [ f_{\xi}( W^{t})] \circ \nabla E_{\xi\sim D} [ f_{\xi} ( W^{t}) ]$$

should instead be
  $$E_{\xi\sim D} t[ \nabla f_{\xi} ( W^{t} )   \circ \nabla f_{\xi}( W^{t} ) ].$$

**Reference:**

Attouch, Hedy, Jérôme Bolte, and Benar Fux Svaiter. "Convergence of descent methods for semi-algebraic and tame problems: proximal algorithms, forward–backward splitting, and regularized Gauss–Seidel methods." Mathematical Programming 137.1 (2013): 91-129.

---

### Decision · Program_Chairs · 2024-09-25

**Decision:**

Accept (poster)

**Comment:**

All reviewers reach a consensus that the paper is worthy of publication. I suggest the authors to incorporate their general responses and necessary replies during the discussion to the revision of the paper.